# Prediction of transition state structures of gas-phase chemical reactions via machine learning

Sunghwan Choi [1] ✉

The elucidation of transition state (TS) structures is essential for understanding the mechanisms of chemical reactions and exploring reaction networks. Despite significant advances in computational approaches, TS searching remains a challenging problem owing to the difficulty of constructing an initial structure and heavy computational costs. In this paper, a machine learning (ML) model for predicting the TS structures of general organic reactions is proposed. The proposed model derives the interatomic distances of a TS structure from atomic pair features reflecting reactant, product, and linearly interpolated structures. The model exhibits excellent accuracy, particularly for atomic pairs in which bond formation or breakage occurs. The predicted TS structures yield a high success ratio (93.8%) for quantum chemical saddle point optimizations, and 88.8% of the optimization results have energy errors of less than 0.1 kcal mol⁻¹. Additionally, as a proof of concept, the exploration of multiple reaction paths of an organic reaction is demonstrated based on ML inferences. I envision that the proposed approach will aid in the construction of initial geometries for TS optimization and reaction path exploration.

Transition states (TSs) are essential for understanding chemical reactions and reaction networks[1–3]. In principle, the characteristics of chemical reactions are determined not by a specific molecular conformation but by molecular trajectories that are affected by the entire potential energy surface (PES). Various advanced computational methods have been proposed to simulate the molecular trajectories of chemical reactions effectively[4–8]. However, owing to the tremendous computational costs of obtaining accurate PESs and trajectories, the elucidation of chemical reactions still relies heavily on classical TS theory[9,10]. TS theory is used to derive many reaction properties from a TS, such as saddle points on PESs[11,12].

Experimentally capturing TS structures is impractical owing to their transient nature. However, they can be accurately derived using computational methods. Over the past five decades, various quantum chemical methods using the first and second derivatives of energy with respect to atomic coordinates have been proposed to capture saddle points in PESs accurately[13–16]. These methods can be used to determine the TS of a target reaction reliably based on a given initial structure. However, they require precise chemical knowledge regarding a target reaction to construct an initial conformation. Based on the high complexity of PESs and high sensitivity of quantum chemical calculations to initial structures, dozens of numerical experiments may be required to design an appropriate initial structure, even for trained chemists. This repetitive trial-and-error process is one of the major hurdles for calculating TS structures, along with the high computational cost of quantum chemical methods.

Machine learning (ML) techniques have been broadly applied in many areas of chemistry to derive chemical knowledge from a wide database[17–19]. Based on the introduction of various neural network architectures, many molecular and reaction properties (e.g., molecular energy and activation barriers) can be predicted accurately[20–23]. Despite significant advances in ML techniques, the prediction of 3D

[1]Division of National Supercomputing, Korea Institute of Science and Technology Information, 245 Daehak-ro, Yuseong-gu 34141 Daejeon, Republic of Korea.
✉e-mail: sunghwanchoi@kisti.re.kr

molecular structures is not easily achievable because inferred 3D structures must satisfy not only permutation invariance with respect to atom order, but also physical (rotational and translational invariances) and Euclidean (triangle inequality) constraints.

Recently, various researchers have developed ML architectures that can predict TS structures while satisfying these conditions. Pattanaik et al[24]. used a graph neural network to generate an initial interatomic distance and weight matrix from the interatomic distances of reactant and product structures. This model, which is referred to as TSGen hereinafter, adopts internal nonlinear optimization to find the atomic positions with the interatomic distance matrix closest to the initial matrix. A second model called TSNet was proposed by Jackson et al[25]. This model is based on a tensor-field network that applies spherical harmonics as convolution filters to distinguish relative atomic positions and directly predicts the atomic positions of TS structures[26]. These ML architectures mathematically satisfy the conditions for determining TS structures. To achieve further improvements in terms of the quality of prediction, specialized ML models for specific reaction types can be introduced[27]. However, the applicable ranges of specialized ML models are limited to reactions with the same mechanisms. Therefore, these models are unsuitable for exploring the full chemical reaction space.

In this paper, an ML model that infers the TS structures of general single-step reactions is proposed and used to find multiple TSs that yield the target reactants and products. By using two organic reaction databases, it is confirmed that the proposed model outperforms existing models, particularly for predicting rarely distributed cases (e.g., bond formation or breakage). In addition to estimating the errors of predicted distances, quantum chemical calculations are performed to measure the chemical validity of predicted TS structures. The most frequently predicted TS structures converge to saddle points, and their energies have strong agreement with the reference energies. Additionally, based on ML predictions of TS structures, a fast TS finder is implemented.

## Results

### Prediction accuracy

Schematic representations of the proposed model architecture and its inference procedure are presented in Fig. 1. The proposed model is designed to predict the interatomic distances of TS structures based on three molecular structures, namely reactants, products, and their linear interpolations. Each structure is represented by a set of atomic pair features. The pair features of the three structures are constructed by concatenating features from two atomic numbers and an interatomic distance, $d_{ij}$. The sets of pair features from reactant, product, and linearly interpolated structures are marked with green, red, and purple round boxes. These sets of pair features are updated by pair sequence interaction (PSI) layers without loss of permutation invariance and size extensivity. Figure 1b presents a PSI layer consisting of two different types of components, namely transformer encoder and bidirectional gated recurrent unit (GRU) layers. The transformer encoder updates pair features by reflecting only the set of pair features belonging to one structure, whereas the following bidirectional GRU updates feature by considering the same pairs in different structures. Based on the updated features of the interpolated structures, $f_{ij}^{\mathrm{I,pair}}$), the model predicts the normalized distances of TS structures based on those in the interpolated structure, $d_{ij}^{\mathrm{TS}}/d_{ij}^{\mathrm{I}}$. Additional details regarding the model architecture and training procedure are provided in "Model architecture and training".

The model outputs are then converted into the predicted interatomic distances, $d_{ij}^{\mathrm{TS}}$, and used to generate the atomic positions in TS structures, $X$, through nonlinear optimization. As shown in Fig. 1c, the nonlinear optimization process determines atomic positions by minimizing the differences between predicted and reconstructed interatomic distances. To mitigate the effects of prediction errors on the

results of nonlinear optimization, inferences from multiple trained ML models can be utilized to form an ensemble. In this study, 90 trained models were employed for ensemble predictions. For both single-model and ensemble cases, linearly interpolated structures are used to determine the initial geometry for nonlinear optimization. The details of ensemble prediction are described in "Nonlinear optimization and ensembles".

The proposed model was trained and validated using the organic reaction database released by Grambow et al[28]. Table 1 presents the test errors of the single-model and ensemble prediction results in terms of two metrics: molecular mean absolute error (MAE) and molecular mean absolute percentage error (MAPE). Molecular MAE is appropriate for estimating the overall deviations of bond distances. However, from a chemical perspective, errors in interatomic distances are not fairly important. While relatively large errors can be tolerated at sufficiently long interatomic distances, even small errors can be critical to a TS structure if they occur in the chemical bonding region. To reflect errors at small distances sensitively, the molecular MAPE is used in conjunction with the molecular MAE. The mathematical definitions of both metrics are provided in "Metrics".

For the test subset, the molecular MAPEs of the single-model and ensemble were measured as 3.681% and 3.407%, respectively. The corresponding molecular MAE values are 11.56 pm and 10.70 pm. The error is reduced further by test-time augmentation (TTA), which utilizes the results of inferences of augmented test inputs to mitigate the variance of test inferences. For image data, flipped, rotated, and translated test images were used to enhance the quality of predictions. TTA can be implemented in many different ways depending on the methods used to augment data and merge inferences[29,30]. In this study, augmented data were obtained by reversing the directions of chemical reactions, and the predicted interatomic distances from both original (forward) and reversed (backward) reactions are averaged. This not only enhances accuracy, but also eliminates the directional dependence of TS structures, which is an important invariance. Because this augmentation was not applied during training, no problems associated with artificial data such as reduced generalization were introduced.

The interatomic distances obtained from ML inferences are highly accurate. However, they do not directly correspond to reliable 3D TS structures. Fortunately, based on nonlinear optimization using the results of inferences, accurate molecular geometries whose errors in terms of interatomic distances are less than those of both single-model and ensemble results can be obtained. This indicates that the remaining error in the predicted distances can be mitigated by constraining the set of interatomic distances to satisfy the Euclidean condition. Despite highly accurate results using nonlinear optimization, because enantiomers are not distinguishable in terms of interatomic distances, nonlinear optimization cannot guarantee the correct chirality of TS structures. The incorrect prediction of chirality is not considered by the error metrics adopted in this study. Therefore, incorrect chirality prediction is observed even in the lowest-error case (0.88% molecular MAPE and 2.28 pm molecular MAE) shown in Supplementary Figure 1, which plots a few best and worst prediction results. This chirality issue is a common limitation of the ML model based on interatomic distance [24].

To evaluate our model in comparison to other ML models, two existing models, TSGen and TSNet, were trained on the same train dataset[24,25]. . TSGen exhibited slightly better performance than TSNet. However, both models exhibited more than twofold greater errors (for both error metrics) compared to the proposed model.

For detailed error analysis, Fig. 2 plots the averages of the absolute percentage errors for four different categories of atomic pairs. The first category, which is the most common in the database, corresponds to atomic pairs that are not bonded in either the reactants or products. The second and third categories correspond to atomic pairs that undergo bond formation and breakage, respectively. The last

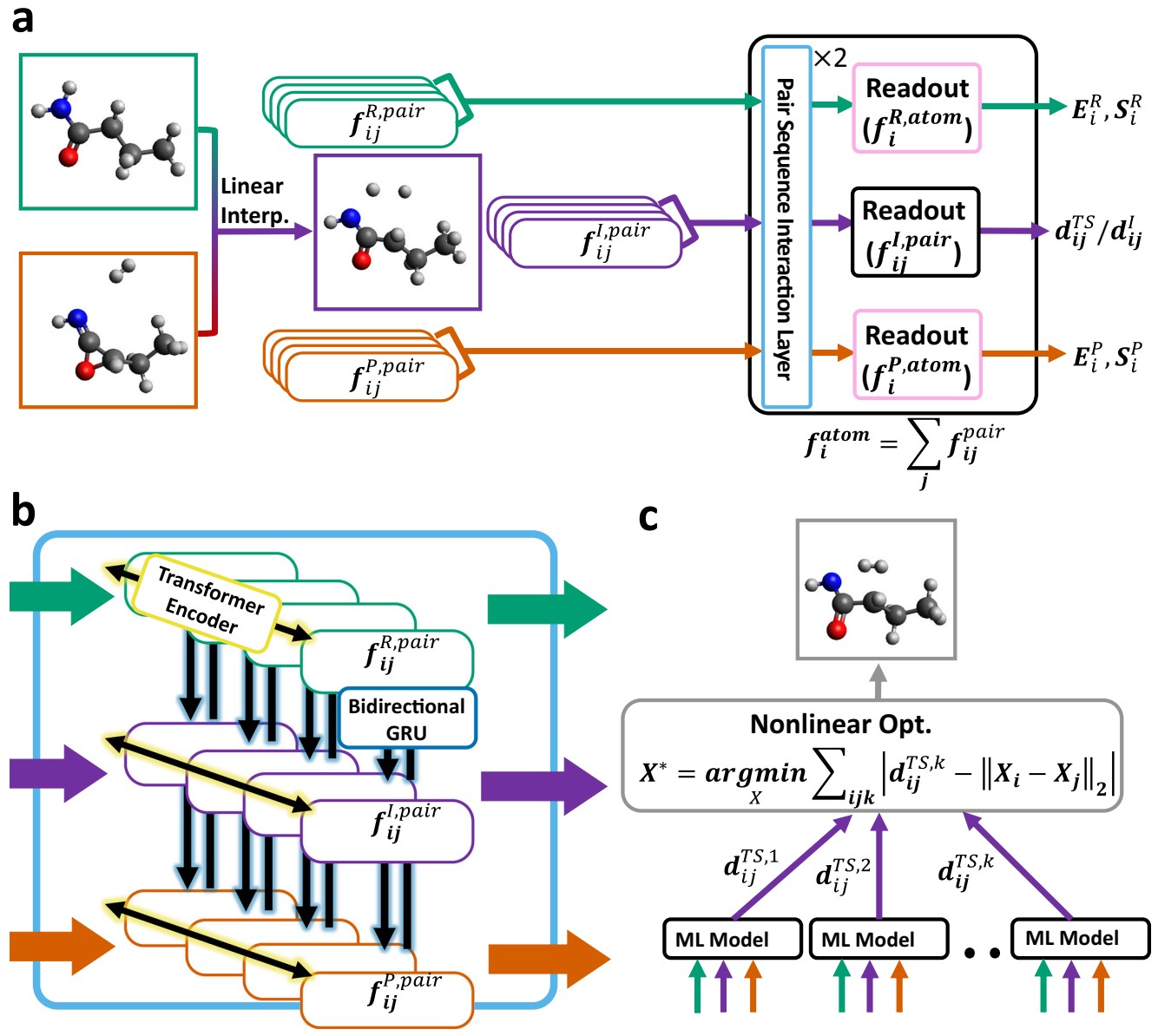

**Fig. 1 | Schematic representations of machine learning (ML) models. a** Overall ML procedure utilizing reactant, product, and linearly interpolated structures. Two different readout layers derive the molecular properties (molecular energy and entropy values, $E$ and $S$) and the ratio between interatomic distances of linearly interpolated structures, $d_{ij}^{I}$, and transition state, $d_{ij}^{TS}$, from the pair features of three structures ($f_{ij}^{R,pair}$: reactant, $f_{ij}^{I,pair}$: linearly interpolated structure, $f_{ij}^{P,pair}$: product). **b** Illustration of the pair sequence interaction layer consisting of transformer encoder and bidirectional gated recurrent unit(GRU), which are responsible for intermolecular interaction among three structures and interatomic interactions in each structure, respectively. **c** Predicted interatomic distances are used to reconstruct 3D atomic positions, $X$, through nonlinear optimization. For ensemble predictions, the results from multiple models can be utilized to perform a single nonlinear optimization.

category corresponds to atomic pairs that are bonded in both the reactants and products. Because only a limited number of bond changes occur in single-step reactions, the second and third cases are relatively rare (1.27% and 1.96% of all interatomic pairs in the training set, respectively). Despite the limited number of the pairs belonging to the second and third categories, the ensemble predictions yields ~5% errors in both categories, whereas the two comparison models yield errors of >15%. For the prediction of TS structures, determining

interatomic distances that induce chemical bond formation and breakage is essential. The comparison models fail to identify such distances accurately.

To highlight the dependency of model accuracy on interatomic distances and atomic numbers, the changes in absolute percentage errors and distributions of atomic pairs in the database are presented in Fig. 2b. The colored bars represent the distributions of six different combinations of atomic pairs at intervals of 4.125 pm. For improved

readability, Fig. 2b visualizes six cases among 10 possible combinations. The entire distribution is presented in Supplementary Figure 2. The dotted, dashed, and solid black lines represent the absolute percentage errors of the interatomic distances of TSNet, TSGen, and the ensemble approach. The points along the presented lines were derived by averaging the percentage errors of all training interatomic distances belonging to the same interval. The red line represents the criterion for chemical bonding at a value of 156.6 pm. In Fig. 2b, relatively large absolute percentage errors can be commonly observed for interatomic distances where the data are rarely distributed. In particular, in the bonding region (under 156 pm), all models yield relatively high errors, except around the 105 pm and 150 pm regions, where large numbers of C–H and C–C bonds are distributed. Despite this common trend, the proposed model outperforms both TSNet and TSGen over the entire distance range. Even for atomic pairs in the bonding region, maximum errors of 6% can be observed for the ensemble prediction approach, whereas the comparison models record up to 15% or more errors. This analysis confirms that the proposed model provides reliable predictions for all types of atomic pairs even for atomic pairs that have chemical bonds. Additionally, it provides relatively high accuracy for atomic pairs consisting of infrequently distributed elements. All inference results of TSNet, TS.

To investigate the applicability of the proposed approach to a small database, model performances trained with $S_N2$ reaction

database and reduced Grambow's database are measured. The $S_N2$ reaction database was published by the authors of TSNet[25]. For $S_N2$ reactions, the proposed ensemble approach yielded a molecular MAPE of 1.738% and molecular MAE of 4.54 pm. The best-reported MAE for the TSNet model is 18.31 pm[25]. The definition of error in the TSNet model study is slightly different from the molecular MAE. If the same definition to the TSNet results is applied, then the error of the ensemble prediction is 4.97 pm. These results indicate that the proposed model is more than three times as accurate as TSNet. Although the $S_N2$ database contains only 48 training data, the proposed ML approach still yields high accuracy. Additionally, for the tests based on the reduced training set of Grambow's reaction database, the proposed model achieved high accuracy. The accuracy of ML prediction according to the size of the training set is presented in the Supplementary Figure 3 of the Supporting Information. The proposed model trained with only 25% of randomly sampled reactions from the training set outperforms the comparison models trained with the entire training set. These test results verify that the proposed ML architecture can learn the interatomic distances of TS structures efficiently, even with a small number of reaction data.

To estimate the chemical validity of the predicted TS structures, quantum chemical simulations were performed. Although the $S_N2$ database contains a number of realistic chemical reactions, validation based on quantum chemical calculations was conducted only for the database released by Grambow et al[28]. because of the small size of the $S_N2$ database.

### Quantum chemical validation

To validate prediction quality, saddle point optimizations were conducted using the predicted TS structures as initial structures. Among the 1196 test molecules, 1122 (93.8%) of the molecular structures successfully converged. Among the failed saddle point optimizations, 60 failed because the maximum number of iterations for geometry relaxation was exceeded, six failed because the self-consistent field failed to converge, and eight failed for other reasons. Unlike the reference calculations, the saddle point calculations performed in this work used ML inferences as initial structures. Although all options of saddle point optimization except for the initial structure used the same values as the reference options, saddle point calculation may fail to converge or converge to the different structure to the reference

### Table 1 | Accuracy of trained models for predicting interatomic distances

| Prediction types | MAPE (%) | | MAE (pm) | |
|---|---|---|---|---|
| | Single model | Ensemble | Single model | Ensemble |
| Prediction without TTA | 3.681 | 3.407 | 11.56 | 10.70 |
| Prediction with TTA | 3.642 | 3.404 | 11.44 | 10.69 |
| Prediction with TTA and NLOpt | 3.365 | 3.083 | 10.29 | 9.53 |
| TSGen[24]/TSNet[25] | 7.738/9.229 | –/– | 22.46/24.37 | –/– |

Molecular mean absolute percentage error (MAPE) and molecular mean absolute error (MAE) of single-model, ensemble, and comparison model results. The machine learning predictions are performed under different conditions (with or without test-time augmentation (TTA) and subsequent nonlinear optimization (NLOpt)). The last row lists the test errors of ML models proposed by other groups. All compared models were trained on the same database used for training the proposed model.

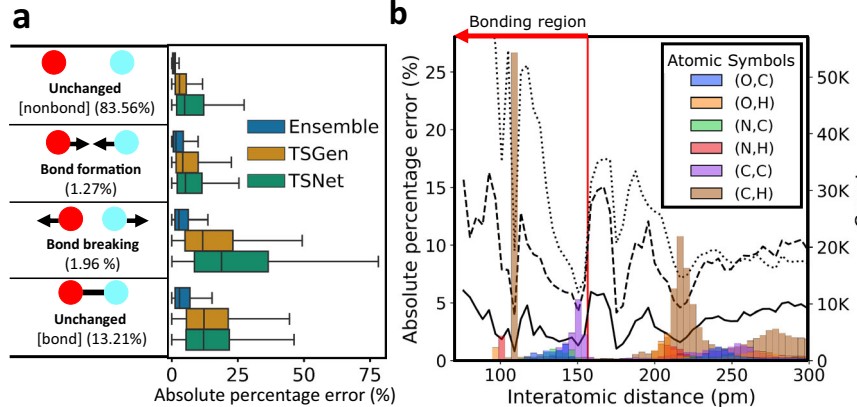

**Fig. 2 | Error analysis of ML predictions. a** Boxplot of absolute percentage errors of four interatomic pairs depending on bond change types during a reaction. Blue, orange, and green boxes represent the prediction errors of TSNet, TSGen, and the ensemble approach, respectively. The percentages in parentheses represent the ratio of the type of interatomic pair among the entire training set. The presence or absence of chemical bonds in reactants and products is simplified as determining whether the interatomic distance is <156.6 pm. **b** Absolute percentage errors and numbers of atomic pairs with different element sets according to interatomic distance. The

dotted, dashed, and solid lines represent the average errors of TSNet, TSGen, and the ensemble approach, respectively. The bars represent the frequencies of distances in the training set by atomic number. The red line represents a value of 156.6 pm, which is the criterion for the presence of a chemical bond. For clarity, only the distributions of selected atomic pairs are visualized. The distributions of the atomic pairs that are not visualized are plotted in Supplementary Figure 2 in the supplementary information. All predicted interatomic distances from the TSNet, TSGen, and Ensemble methods are provided in Supplementary Data 1–3, respectively.

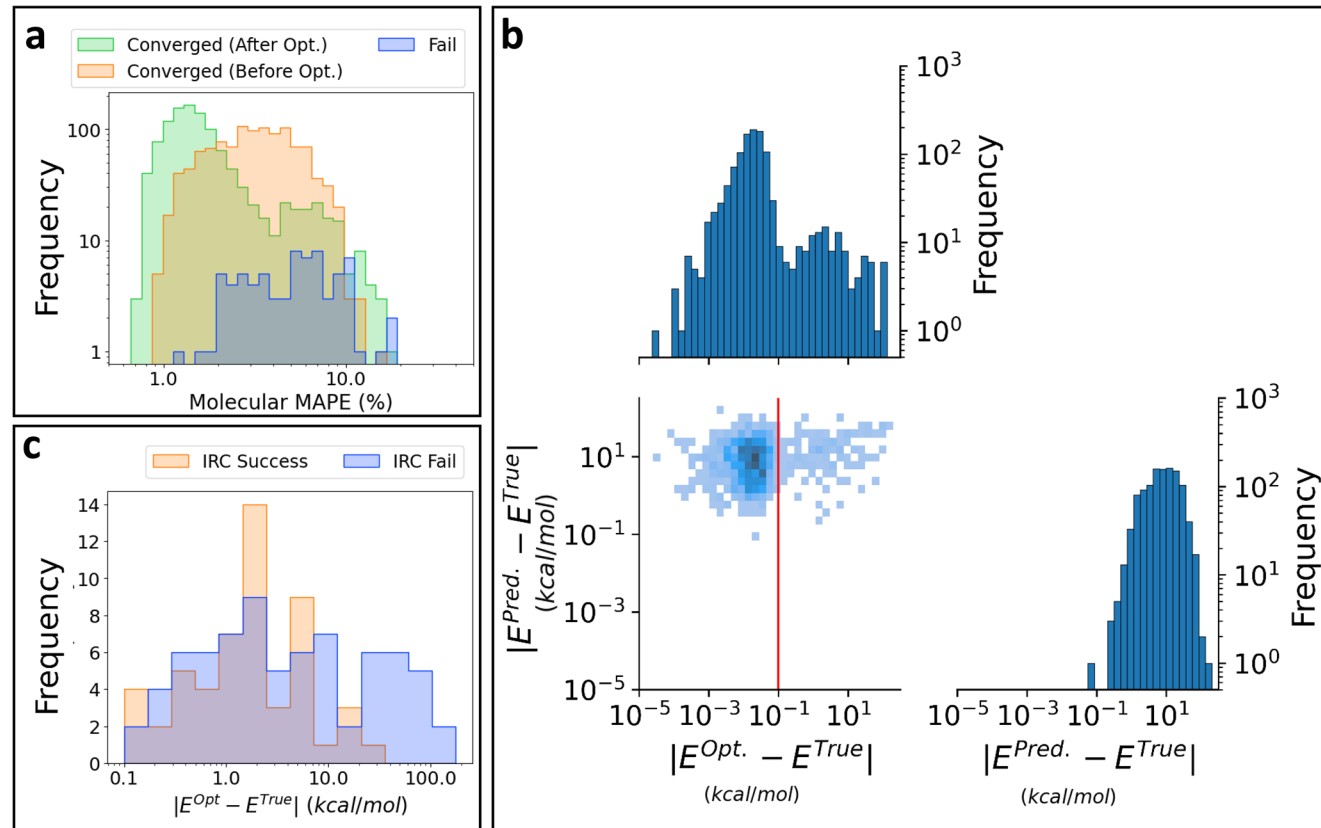

**Fig. 3 | Quantum chemical validation results for predicted transition state structures. a** Distributions of molecular mean absolute percentage error (MAPE) values of inferred and quantum chemically optimized TS structures. Orange and blue bars indicate the molecular MAPE values of structures that yielded successful and unsuccessful convergence in the saddle point optimizations, respectively. Green bars indicate the molecular MAPE values of optimized structures.

**b** Distributions of absolute energy errors of inferred and optimized structures. The red line represents an absolute energy error of 0.1 kcal mol⁻¹. Intrinsic reaction coordinate (IRC) calculations were performed for the structures on the right side of the red bar (126 cases). **c** Distributions of absolute energy errors of structures whose reference products and reactant structures were acquired (orange) or not acquired (blue) through IRC calculations.

one. (detailed options for quantum chemical calculation parameters are provided in "Data preparation and quantum chemical validation") Without any manual processing of initial structures, a high convergence ratio for saddle point optimizations was still obtained. Additionally, it does not necessarily indicate that the ML model failed if the initial structures from the model do not converge because convergence is affected not only by initial structures but also by various parameters and methods of optimization.

For the converged structures, frequency calculations were performed and it was observed that 956 structures (80% of the 1196 test set data) had one negative frequency. The high success ratio is not direct evidence of the high accuracy of the ML inferences but considering the fact that Grambow's reaction database includes many non-trivial reactions, an 80% success ratio without any manual handling is a noteworthy achievement that reflects the fidelity of ML inferences as initial structures for saddle point calculations.

The effects of saddle point optimization on TS structures are visualized in Fig. 3. The orange and green bars in Fig. 3a represent the molecular MAPE values of the initial and final structures for saddle point optimization, respectively. The MAPE distribution shifts to the left by the saddle point optimization, which corresponds to a reduction of the error. However, a considerable portion of the structures remains at ~8% molecular MAPE, which yields a bimodal distribution. Splitting of the distribution caused by quantum chemical refinement can also be observed in the energy error. Figure 3b presents the distribution of the absolute energy error for the optimization results. The absolute energy errors of the predicted structures are broadly distributed from 0.1 to 100 kcal mol⁻¹. However, for the optimized

structures, the absolute energy errors of most structures (996 cases) are reduced to below 0.1 kcal mol⁻¹ and only a small number of cases remain within the original error range. This has two major implications. First, the energy of the inferred structures is not sufficiently accurate, even if the inferred interatomic distances have an error of only a few percent on average. Second, quantum chemical optimization reduces energy errors and yields reliable energy in most cases. However, in some cases, the results of saddle point optimization are still far from the reference TS.

In addition to frequency calculations, intrinsic reaction coordination (IRC) calculations were performed to investigate the validity of the optimized TS structures. The structures with errors >0.1 kcal mol⁻¹ (shown on the right side of the red line in Fig. 3b) were selected for testing (126 cases). In principle, a TS structure has a single negative frequency and an IRC calculation beginning from a TS structure provides target reactants and products. However, among the 126 converged TS structures, there were only 45 structures that satisfied both conditions (8 and 58 structures did not satisfy the negative frequency and IRC conditions, respectively, and 15 structures failed on both criteria). If the reference reactant and product structures were obtained by IRC calculations, then the cases were classified into the IRC success group. Otherwise, they were classified into the IRC failure group. The overall distributions of both groups according to the energy error ($|E^{Opt} - E^{Ref}|$) are represented in Fig. 3c. Additionally, each distribution of structures having a different number of negative frequencies is plotted in Supplementary Figure 4.

Generally, the IRC success cases (53 structures) have lower $|E^{Opt} - E^{Ref}|$ values than the IRC failure cases. However, wide overlap

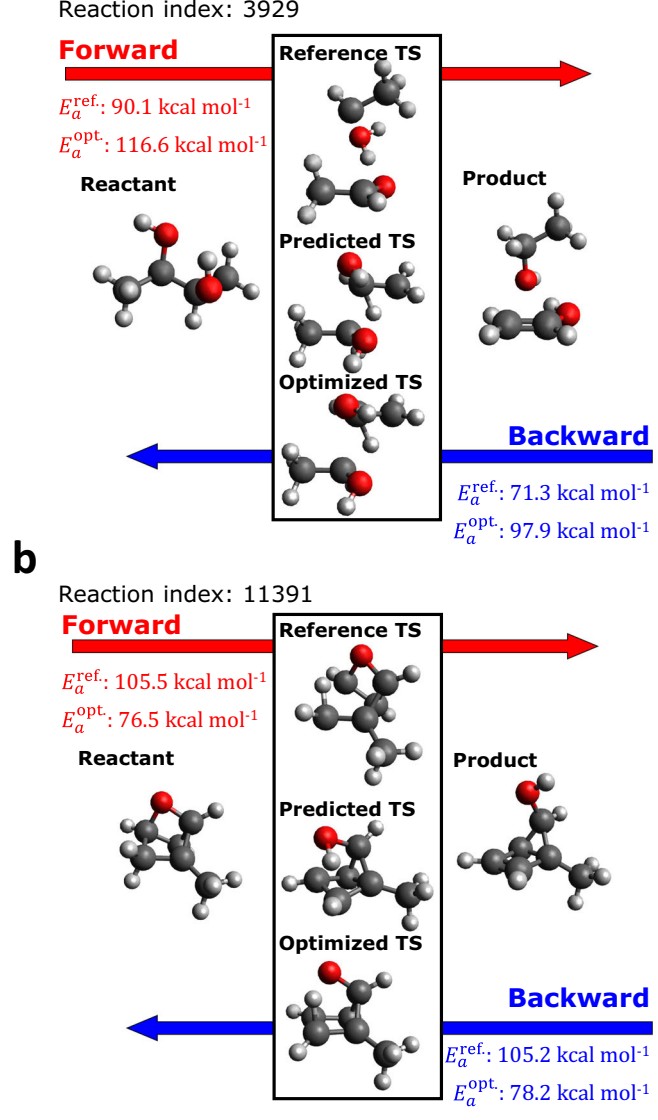

**a**

Reaction index: 3929

**Forward**

$E_a^{\text{ref.}}$: 90.1 kcal mol$^{-1}$

$E_a^{\text{opt.}}$: 116.6 kcal mol$^{-1}$

**Reactant**

**Reference TS**

**Predicted TS**

**Optimized TS**

**Product**

**Backward**

$E_a^{\text{ref.}}$: 71.3 kcal mol$^{-1}$

$E_a^{\text{opt.}}$: 97.9 kcal mol$^{-1}$

**b**

Reaction index: 11391

**Forward**

$E_a^{\text{ref.}}$: 105.5 kcal mol$^{-1}$

$E_a^{\text{opt.}}$: 76.5 kcal mol$^{-1}$

**Reactant**

**Reference TS**

**Predicted TS**

**Optimized TS**

**Product**

**Backward**

$E_a^{\text{ref.}}$: 105.2 kcal mol$^{-1}$

$E_a^{\text{opt.}}$: 78.2 kcal mol$^{-1}$

**Fig. 4 | Molecular structures and activation barriers for extreme cases.** Reactant, product, and transition states having **a** the most positive and **b** most-negative energy errors among the inferred structures that yielded correct reactant and product structures through intrinsic reaction coordinate calculations. A positive (or negative) error indicates that the obtained transition state structure is energetically less (or more) favorable than the corresponding reference transition state.

between the two distributions is observed, indicating that a low energy error does not guarantee the chemical validity of a TS structure. As shown in Fig. 3c, even some TS structures having errors <0.2 kcal mol$^{-1}$ do not provide the desired reactants and products. This phenomenon occurs even in manual saddle point calculations and is one of the major bottlenecks in the TS searching problem. Interestingly, in contrast to the failed cases with small errors, a few TS structures having errors greater than 10 kcal mol$^{-1}$ successfully recover reactants and products. This indicates that the optimized TS structures have a significantly different conformation compared to the reference structure because of the non-uniqueness of reaction paths.

Figure 4 presents two extreme IRC success cases in terms of their energy errors. The upper and lower subplots show the reference, predicted, and optimized TS structures for the cases with the most positive and most-negative energy errors, respectively. More examples

of large positive and negative error cases are plotted in Supplementary Figs. 5 and 6, respectively. In Fig. 4 as well as both Supplementary Figs., one can see that the predicted TS structures are not significantly changed by saddle point optimization, whereas the reference TS structures are noticeably different from both the predicted and optimized TS structures. This means that the proposed model can predict TS structures that are well-optimized, but different from the reference structure. In the positive error case, the activation energy of the reaction passing through the reference TS ($E_a^{\text{ref}}$) is lower than that of the optimized TS ($E_a^{\text{opt}}$) because the energy of the optimized structure is higher than that of the reference structure. In contrast, a negative error indicates that $E_a^{\text{opt}}$ is less than $E_a^{\text{ref}}$, meaning the ML method predicts a TS structure that is more energetically favorable than the reference structure.

Multiple TSs for a chemical reaction is frequently observed in general polyatomic systems. In a strict sense, it is extremely difficult to target the most stable TS structure precisely. However, practically, the most stable TS structure can be found by exploring multiple TSs. Therefore, the generation of multiple reaction pathways is essential for determining the most favorable reaction pathways. Furthermore, to design catalysts or retrosynthetic pathways, the exploration of many competitive reactions is frequently required[31–33]. To tackle this problem, the generation of multiple TS structures using fast and accurate ML inference is demonstrated with the aid of reactant and product sampling.

## Exploring multiple reaction paths

For the reactions on which the ML model failed the most (Fig. 4a), multiple TS structures were derived by using an ensemble approach with normal mode sampling (NMS) for reactants and products. NMS can quickly sample a large number of thermally activated structures because it perturbs atomic positions along the normal modes of equilibrium structures instead of performing heavy molecular dynamic simulations[34]. From the 2000 sets of sampled reactants and products, the same number of TS structures was generated using this ensemble method. Generating 2000 TS structures typically requires significant human labor or computational resources, even if an automatic reaction pathfinder is applied. However, only one minute is required for inferring 2000 TS structures using an NVIDIA Titan RTX graphic card and the ensemble method. Although 90 ML models were used for ensemble predictions, inferences can be performed in parallel. Therefore, the computational expense of ensemble inferences is manageable. To reduce the computational costs of quantum chemical refinement, instead of simulating all generated TS structures, 117 representative TS structures were selected via clustering and then used for saddle point calculations as initial geometries (see "Normal mode sampling and clustering" for the details of the NMS and clustering procedures). Among these 117 calculations, 24 calculations successfully converged.

Figure 5 presents the relative energies ($|E^{\text{TS,opt}} - E^{\text{TS,ref}}|$) of the 24 successfully converged TS structures relative to the reference structures. Among the 24 structures, four conformations can be observed. The energy deviations among structures with the same conformations are >0.05 kcal mol$^{-1}$. The energies of each conformation and corresponding structures are presented in the same color. The conformation with the lowest energy (red box) is the same as that of the reference TS. The conformations in the blue and pink boxes share most structural characteristics, except for the direction of the −OH group. In contrast, the conformation in the green box has a completely different structure, even though its energy is close to those of the other two conformations. By utilizing ensemble inferences with NMS, four different TS structures can be obtained within a reasonable number of quantum chemical calculations.

To validate the TS structures, IRC calculations were performed on the four conformations shown in Fig. 5. It is confirmed that the

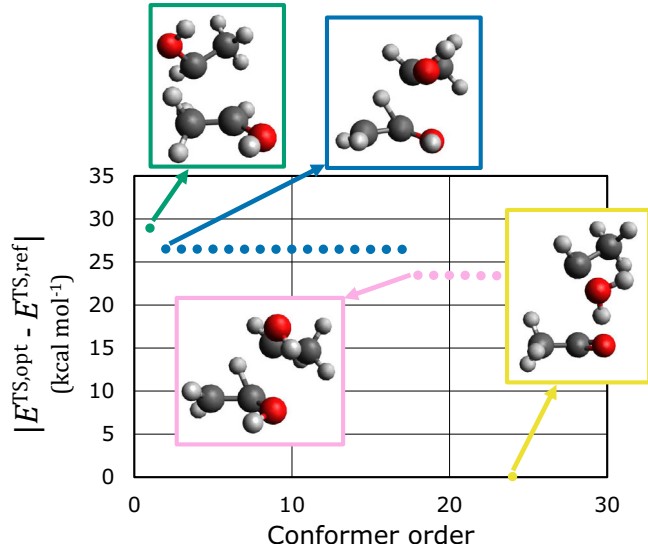

**Fig. 5 | Relative energies derived through transition state (TS) exploration.** TS exploration was performed for the reaction presented in Fig. 4a using quantum chemical saddle point optimization and the proposed machine learning approach with off-equilibrium samplings of reactants and products. The relative energies of 24 TS structures from the exploration are plotted, in descending order with respect to the energy of the reference TS structure ($E^{TS,opt} - E^{TS,ref}$). The energies with the same color share the same conformation. The representative structures are presented in boxes of the same color.

reference reactant and product were obtained from IRC calculations using the structures shown in green, blue, and pink boxes as initial geometries. In contrast, the structure in the yellow box, which has the same conformation as the reference TS, does not yield the correct product structure. Instead of the reference product, this IRC calculation yields a conformation consisting of three molecules ($C_2H_5$, $H_2O$, and $C_2OH_3$). Interestingly, the same products are generated through IRC validation of the reference TS structure. Because the reference TS structures were generated using the single-ended growing string method followed by refinement using saddle point optimization, the obtained TSs can be transformed into the TSs of unwanted reactions during saddle point optimization. Although one of the obtained TSs is not a correct TS for the target reaction, this result confirms that the proposed method can generate reliable TS structures, even with off-equilibrium reactant and product structures. Additionally, the effectiveness of the proposed approach for tackling multiple reaction paths has also been confirmed.

The IRC validations of the obtained TS conformations reveal that the large energy errors of the originally predicted TS structures may not be actual errors of the model, but of the reference database. To evaluate data errors in the TS database systematically, a large number of IRC calculations are required, which is not viable with available computational resources. Despite some potential faults in the reference structures, the proposed ML model was trained stably and can be applied to exploring multiple reaction paths, which is a challenging problem in many contexts of chemistry.

## Discussion

In this study, an ML model for the fast and accurate prediction of TS structures was proposed and validated using general organic chemical reactions. To achieve high accuracy with the satisfaction of various necessary invariances, the ML model utilizes PSI layers designed to handle any size of molecules and any type of element. These layers do not rely on any prior knowledge regarding chemical reaction types, but on reactant and product structures, so the proposed model can be extended to all types of chemical reactions without modifications.

Despite a lack of reliable TS data, the proposed ML model exhibits high accuracy for general organic reactions. In particular, for atomic pairs that undergo bond formation or breakage during reactions, the proposed model significantly outperforms existing models. Additionally, when the inferred TS structures are used as inputs for quantum chemical saddle point optimizations, a high convergence ratio is achieved, which demonstrates that the proposed ML method can serve as an attractive solution for initializing a starting structure for saddle point optimization.

To demonstrate the potential usage of the proposed model for searching multiple reaction paths, an autonomous reaction pathfinder was implemented and tested on a polyatomic reaction. By performing ML inference on the off-equilibrium conformations of reactants and products, and further refinement through saddle point calculations, various chemically meaningful TS structures were identified. This automated TS finder has huge potential to leverage ML applications for chemical reactions because it can be directly utilized for not only chemical applications (e.g., design of synthetic routes and catalysts), but also various ML applications related to chemical reactions (e.g., active learning). Although off-equilibrium conformations are utilized to predict TS structures, they are still sampled from well-optimized and aligned structures, which are obtained from quantum chemical calculations. This process is not trivial, particularly for bimolecular and trimolecular reactions, because the initial relative pose of molecules is a critical factor in determining reaction profiles. Therefore, to develop a more rigorous and realistic strategy for TS structure prediction, combinations of various ML approaches presented in this work (e.g., TTA and ensemble), as well as high-end quantum chemical calculation methods for configuration sampling are required.

## Methods

### Data preparation and quantum chemical validation

In this study, two publicly available reaction databases were used. The first was released by Grambow et al[28]. This database consists of density functional theory (DFT) results with two different options, namely B97-D3/def2-mSVP and $\omega$B97X-D3/def2-TZVP. Here, $\omega$B97X-D3/def2-TZVP results (11961 reactions) were adopted based on their reliable accuracy. To extract atomic positions and molecular properties from DFT results, the *ard_gsm* package provided by the authors of the database was utilized (www.github.com/cgrambow/ard_gsm). The database was split randomly into training, validation, and test sets with an 80-10-10 ratio. The second database contains 53 $S_N2$ reactions and was released by the author of TSNet[25]. This database includes five validation and 48 training data. The same splitting result is adopted.

Random sampling is the simplest method for splitting a database into train, validation, and test subsets. However, in a reaction database, random sampling may lead to the same structures being included in both the training and test sets because reactions sharing a reactant or product are treated as different reactions in a reaction database. To avoid duplicates of structures, in some ML applications for chemical reactions, reactant-based scaffold splitting, which assigns reactants having the same scaffold to the same subset, is employed[35,36]. Supplementary Table 1 summarizes the accuracy of ensemble predictions trained using Grambow's database with two different sampling methods. The results indicate that the proposed model provides reliable accuracy, regardless of the splitting method. Therefore, only random sampling was used for further quantum chemical validations.

Saddle point optimization and IRC calculations were performed on the test reactions in the first database using the same DFT functional ($\omega$B97X-D3) and basis set (def2-TZVP). Saddle point optimizations were performed with a finite difference Davidson method implemented in QChem, whereas the Pysisyphus program was utilized for IRC calculations[37]. The Pysisyphus program provides various methods for exploring PESs based on the results (mainly gradients and Hessian) of a quantum chemical package. To execute IRC calculations

using Pysisphus, the Psi4 package was employed because it is seamlessly coupled with Pysisyphus and supports both $\omega$B97X-D3 and def2-TZVP[38]. For both Pysisyphus and the Psi4 package, default options were employed. The saddle point calculations using the QChem package were performed with the same options used to derive the original database results. The maximum number of SCF cycles and optimization operations were both 100. The tolerances for the gradients, displacement, and energy of optimization were $10^{-4}$, $4 \times 10^{-4}$, and $3.3 \times 10^{-7}$ in atomic units, respectively.

## Model architecture and training

Linear interpolated geometries are frequently used for TS search algorithms because of their reliable structures and low costs[13,39]. The proposed model predicts the interatomic distances of the true TS from the reactant, product, and linearly interpolated structures. These three structures are featurized as sequences of atomic pair sets. Paired features are constructed by concatenating one distance with two atomic features. For atomic and distance features, the entity embedding of atomic numbers and a Gaussian kernel are employed, respectively. For the centers of Gaussian functions, points on an equidistant grid are used. The information regarding this grid is presented in Supplementary Table 2. Each structure includes a set of pair features ($f_{ij}$) and reactions are featurized as a sequence of structure feature sets whose dimensions are ($\frac{N_{atom} \times (N_{atom}+1)}{2}$, 3, $N_f$), where $N_{atom}$ and $N_f$ are the number of atoms and feature size for the pairs, respectively. For convenience, in Fig. 1, two atomic indices ($i, j$) are introduced. However, only one index is used for atomic pairs in the actual implementation. The pair features satisfy the translational and rotational invariances because neither atomic numbers nor distances are changed by translation or rotation.

The number of atoms participating in a reaction is not fixed and the order of atoms is not chemically meaningful, whereas the size of the second dimension is always fixed and the order is not interchangeable. Therefore, operations must be flexible in terms of the size and order of the first dimension. To preserve the nature of a data structure, a novel layer called a PSI layer is introduced.

Initially, the PSI layer updates pair features by reflecting other pairs in the same structure. This procedure utilizes a transformer encoder, which is a permutation-invariant and size-extensive operation. This first update is not affected by the pair features of other structures. Subsequently, atomic features are updated again to reflect the information from other structures. Because this second update proceeds only on atomic pairs sharing the same index, the permutation invariance and size extensivity of the first dimension is still preserved. For the second update, a bidirectional GRU is used. Because a GRU layer must update the features of the same atomic pairs in three structures, atom mapping information is required to implement PSI layers. In this work, the atom mapping information contained in the databases was utilized. To apply PSI layers to data that do not have atom mapping information, such data must be obtained preliminary using various atom mapping methods[40,41].

Figure 1b illustrates the two updates of the PSI layer using different colors. The yellow operation operates only on pair features in the same structure, whereas the following blue operation updates pair features by reflecting pairs with the same index, but in other structures. By stacking multiple PSI layers, the effects of all atomic pairs of reactant, product, and linearly interpolated structures can participate in determining each of the final atomic pair features. In the proposed model, two PSI layers are stacked and the initial feature size of atomic pair features (128) is changed to 512 after the two stacked PSI layers because each bidirectional GRU doubles the feature size.

To compute reaction properties from updated atomic pair features, two readout layers are used. The first of these readout layers, which is represented by the purple box in Fig. 1a, is applied to compute the ratios of interatomic distances between the TS structure and linearly interpolated structure. The second readout layer computes the

contribution of each atom to the total energy, vibrational entropy, and rotational entropy, which are denoted as $E_i$, $S_{(vib, i)}$, and $S_{(rot, i)}$, respectively. The atomic features for the second layer are computed through the following equation:

$$f_i^{atom} = \sum_j f_{ij}^{pair}, \tag{1}$$

where $f_i^{atom}$ and $f_{ij}^{pair}$ are the atomic and pair features, respectively. Because the model only computes $f_{ij}^{pair}$ when $i \geq j$, the omitted pair features are recovered using symmetric approximation as $f_{ij}^{pair} = f_{ji}^{pair}$. The predicted values from the second readout layers are utilized for multi-label learning, which is frequently used to improve the quality of ML inferences by deriving desired information from related information[42,43]. However, in this study, additional labels reduced the accuracy of the ML model (see Supplementary Table 3 in the supporting information.) A detailed description of the loss function design is provided in "Loss function".

All data and codes to train models are described in "Data availability" and "Code availability", respectively. The hyperparameters used to train the model are summarized in Supplementary Table 2.

## Metrics

Molecular MAE and molecular MAPE are defined as follows:

$$(\text{Molcular MAE}) = \frac{1}{M} \sum_t^M \frac{2}{N_{atom,t}(N_{atom,t}-1)} \sum_{i<j} |d_{ij}^{t,true} - d_{ij}^{t,pred}|, \tag{2}$$

$$(\text{Molcular MAPE}) = \frac{1}{M} \sum_t^M \frac{2}{N_{atom,t}(N_{atom,t}-1)} \sum_{i<j} \frac{|d_{ij}^{t,true} - d_{ij}^{t,pred}|}{d_{ij}^{t,true}}, \tag{3}$$

where $d_{ij}^{t,true}$ and $d_{ij}^{t,pred}$ are the interatomic distances between the $i$th and $j$th atoms in the geometries for a $t$-indexed reaction. $M$ and $N_{atom,t}$ are the number of reactions in a database and number of atoms in the $t$-indexed reaction, respectively. The distance error used for TSNet is defined in Equation 4 in ref. [25].

## Loss function

During training, the weighted sum of three losses is minimized. The first loss, $L_1$, is the molecular MAE of the predicted TS interatomic distances. This term directly guides the model to predict the target interatomic distances. The second loss, $L_2$, includes the errors of reaction properties from the second readout layers. As described in "Model Architecture and Training", the second readout layer predicts the atomic contributions of three properties of reactant, product, and TS structures. By summing these atomic contributions, the properties of each structure, namely $E(= \sum_i E_i)$, $S_{vib}(= \sum_i S_{(vib, i)})$, and $S_{rot}(= \sum_i S_{(rot, i)})$, can be derived and the $L_2$ is defined as the sum of the MAE in $E$, $S_{vib}$, and $S_{rot}$ for both reactants and products. For training using the $S_N2$ database, $L_2$ consisted of only the energy term because reference entropy values are not available in this database. The final loss, $L_3$, represents the constraints of the Euclidean distance matrix whose elements are the squares of interatomic distances. In addition to the hollow and symmetric conditions, a Euclidean distance matrix should satisfy the eigenvalue condition, which is non-trivial to implement. The 3D Euclidean distance matrices can have at most 5 nonzero eigenvalues and the sum of eigenvalues should be zero[44]. To implement these conditions, $L_3$ is defined as follows:

$$L_3 = \sum_{i=1}^{\min(5,N_{atom})} e_i + \sum_{i=\min(5,N_{atom})}^{N_{atom}} |e_i|, \tag{4}$$

where $e_i$ is $i$th eigenvalue in descending order of magnitude. The first and second terms guide the sum of the first five eigenvalues and the remaining eigenvalues to zero. By penalizing a distance matrix that cannot realize 3D atomic positions, the results of the proposed model are guided to satisfy the Euclidean constraint.

The total training loss, $L$, is defined as

$$L = cL_1 + c'(L_2 + L_3). \qquad (5)$$

$L_2$ and $L_3$ are designed to improve the quality of predictions by imposing Hammond's postulate and the Euclidean constraint. Unfortunately, the training results when $c'$ is zero are better than those for cases where $c'$ is one, meaning a model trained on a single label yields better performance than one trained on multiple labels. The accuracies of ensemble and single-model predictions based on training with all combinations of $c$ and $c'$ are summarized in Supplementary Tables 3 and Supplementary data 4.

### Nonlinear optimization and ensembles

An ML model such as TSNet directly outputs atomic positions satisfying invariance conditions[25]. However, in some studies, including this study, to preserve the constraints of atomic positions, ML models predict structure-dependent quantities (e.g., a distance matrix[24] or Coulomb matrix[45]), and atomic positions are reconstructed through a subsequent nonlinear optimization. In this study, similar to the TSGen model, a distance matrix was used as an optimization target. In TSGen, nonlinear optimization is a component of ML inference, whereas, in this study, nonlinear optimization was performed after the completion of ML inference. Therefore, the results of multiple ML models can be used in a single nonlinear optimization together. From the initial atomic positions, $X$, the nonlinear optimization finds optimal positions, $X'$, to minimize the differences relative to the ML predictions as follows:

$$X^* = \arg \min_X \sum_k \sum_{ij}^{N_{atom}} w_{ij} |d_{ij}^k - |X_i - X_j|_2|, \qquad (6)$$

where $d_{ij}^k$ and $w_{ij}$ are the predicted interatomic distance from the $k$th model in the ensemble and the weight factor, respectively. To increase the contribution from short interatomic distances, weight factors are defined in a Gaussian form, $e^{-\alpha d_{ij}^2}$. By testing many $\alpha$ values on the validation results, an optimal value of $\alpha(=0.2 \text{ Å}^{-2})$ was obtained and used for the test set.

For ensemble predictions, multiple interatomic distance matrices are required prior to nonlinear optimization. Therefore, the computational costs of inferences are linearly dependent on the size of the ensemble, so the proper selection of an ensemble is important. Sets of models that were obtained from training using different seeds, hyperparameters, and training epochs were utilized. Six different combinations of hyperparameters ($c$ and $c'$ in Equation (5)) are used. For each hyperparameter combination, three independent training runs with different random seeds were performed. During each training run, the model parameters in the epoch yielding the top five molecular MAPE values for the validation dataset were saved and used for the ensemble. Ensemble predictions attempt to mitigate the variance of single-model predictions. However, the participation of low-accuracy models in an ensemble may reduce overall ensemble accuracy. The performances of all individual models are summarized in Supplementary Data 4 and Supplementary Table 3 summarizes the prediction accuracies of many different ensembles. For quantum chemical validation, ensemble predictions from 90 trained models were used and one can see that only the ensemble containing models trained using the optimal hyperparameter set ($c = 2000$ and $c' = 0$) outperforms the ensemble with 90 models.

### Normal mode sampling and clustering

For each reactant and product, 2000 geometries were sampled using NMS[46]. The atomic structures of the sampled reactant and product structures were rotated and shifted to minimize the root-mean-squared deviation between pairs of structures. From the obtained 2000 sets of reactant and product structures, TS structures were predicted based on the proposed ensemble inferences. To avoid generating duplicate TS structures, the 117 representative structures were selected using the affinity propagation clustering method implemented in scikit-learn[47,48]. The features used for clustering were interatomic distances.

## Data availability

The inference results of the QM9 database from TSNet, TSGen, and Ensemble approaches are provided in Supplementary Data 1–3, respectively. All input reaction data for train and inference are already publicly available.

## Code availability

The implementation and training of the proposed model were performed using PyTorch Lightning, which provides a high-level interface for PyTorch and is freely available at https://github.com/Lightning-AI/lightning. The implemented model and scripts for training and inference have been uploaded at https://gitlab.com/sunghwan.choi/learnts.

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

## Acknowledgements

The author thanks Y. Kim for the helpful discussion. This work was conducted with computing resource support from KISTI [KSC-2020-CRE-0117].

## Author contributions

S. Choi designed the overall study, performed numerical experiments, and wrote the manuscript.

## Competing interests

The author declares no competing interests.
