## [Peer Review File · Nature Communications]

Prediction of Transition State Structures of Gas-phase Chemical Reactions via Machine LearningREVIEWER COMMENTS

Reviewer #1 (Remarks to the Author):

Reviewer comment :

This paper presents novel machine-learning approach for prediction of transition state structure optimization for non-periodic molecular reactions. And this outperforms previously suggested models which is quite promising. However there are several points which should be addressed prior to publication.

1. This work exploits extensive reaction dataset of Grambow et al. which contains 10k reactions with atomic indices already mapped between reactant and product. If I understood correctly, in Pair sequence interface layer process, permutation-invariance and size-extensivity are preserved in transformer encoder but it seems correct atom mapping of reactant and product seem to be assumed in "Bidirectional GRU" step. However in other many cases, outside of this dataset, this correct atom mapping between reactant and product is not guaranteed to exist and itself is not a trivial problem. Although this model exhibits excellent performance within current dataset as training input, the author may need to address its applicability toward general molecular reactions. Please clarify in the manuscript if this model can be also generally applicable to any elementary reaction or it requires the atom-mapping process as preliminary step.
2. As this is being machine learning approach, its performance/accuracy scales with the size of the training set. Although dataset of Grambow et al. has vast number of reactions, real applications always involve building up their own reaction data and this is even more stipulated by the failure case given in this manuscript. In this work training-validation-test set as splitted into 8-1-1 ratio, but would be interesting to present performance improvement with respect to number of training set (learning curve). Then readers might be able to expect approximated accuracy when trained with this model.
3. Fig 2. should be improved with better visualization. Font is too small and color should be more differentiable.
4. It's hard to follow what actually "test-time augmentation (TTA)" means in current manuscript without reading the references cited. The author might want to elaborate further in the manuscript regarding this part.
5. In Fig. 4. There are success after/before optimization. If author meant success as convergence of saddle-point optimization, how can you confirm its success even before optimization? Also success is ambiguous terminology which can be better expressed with "converged".
6. In Fig.4, there are obviously four colors contained in histogram but only three of them are explained. What are yellow bars? Also (c) contains three colors and only two of them are labeled. Generally font size of the graph label should be larger. Here color should be more contrasting to make it differentiable.
7. In page 10, "challenging but demanding work" is awkward expression.
8. Regarding saddle-point optimization, author should clarify which specific algorithm has been used in Qchem and which convergence criteria has been imposed in this work. (e.g. Minimum force)
9. Author indicates rotational and translation entropy have been used to encode reaction property, but its notations are wrong. I.e. S_{vib} and S_{rot} , which are vibrational and rotational entropy. Also author needs elaboration why only two components of entropy are used instead of three (e.g. vibrational, rotational and translational entropy)
10. There are several typos throughout the manuscript and these have to be revised before

publication.

Reviewer #2 (Remarks to the Author):

Thank you for the opportunity to review the manuscript by Choi regarding the automated identification of transition state (TS) structures using machine learning (ML).

In the interests of transparency, I'm happy to reveal my identity to the author as Prof. Jason Pearson of the University of Prince Edward Island. This is particularly relevant in this case since I am the corresponding author of a paper that was extensively cited in this manuscript. In particular, my group and I have developed parallel technology for identifying TS structures by ML and Choi has rigorously tested our methodology alongside their own.

Overall I find the work to be a compelling case for the use of ML technology in the typical workflow of a computational chemist interested in elucidating mechanistic and kinetic detail of chemical reactions. Though I must admit that I have not checked out the authors freely available code from their GitHub repository, I have read the paper with interest and studied their methodology carefully. Based on this assessment I can say that the work has been soundly prepared and I see no reason to contrast any of the authors conclusions. Furthermore, the prediction of TS structures is an incredibly challenging endeavour (evidenced by a long history of widely varying methods for doing so) yet is critical for a thorough understanding of virtually any chemical process. As such I must recommend that the manuscript be published.

That being said, I am not without a series of more specific comments (both positive and negative), several of which I would require the author to address in a revised manuscript.

1. It is unclear to me how the author plans to handle chirality with their proposed model. As they state (in many places) most clearly on page 16, "... the ML model predicts only interatomic distances". It is, of course, an inconvenient truth that a stereogenic carbon with 4 unique substituents will produce an equivalent distance matrix of all enantiomers. I cannot interpret how the authors subsequent non-linear optimization step would correct this oversight. So, I ask the author to address this. Specifically, can the model succeed in stereochemical TS predictions? And if so, how?

2. Unfortunately the grammar of the manuscript needs a lot of work. I trust that the Springer/Nature editorial team can work with the author to revise. I spotted no less than 20 errors.

3. I was particularly impressed with the authors ensemble approach, which endows the model with the ability to find multiple TS structures between a particular pair of RC and PC. This is a particularly clever solution to the problem of PES complexity.

4. I found it peculiar that exactly 1196 "test molecules" were selected from the Grambow database for further analysis by optimization (page 8). However, the database contains 11961 reactions (page 13). Is this just a strange coincidence or a typo? If only a coincidence, can the author clarify how the 1196 test molecules were selected from the larger 11961 set?

5. Again, I was confused about the statement on page 9 regarding the error in absolute energy of the optimized structures. It isn't clear to me how one measures this error. I understand that a TS is predicted from the ML model and that is a "predicted" structure. Subsequently a user can optimize that predicted structure using a traditional saddle point optimization from first principles. Therefore an "error" can be determined from the difference in energies of these two structures. However, What structure does one compare to if they wish to assess the error of the "optimized" TS structure? Even the reference database used the same theoretical technique (namely, ω B97X-D3/def2-TZVP). I would ask that the author please elaborate on their explanation of "error" for optimized structures.

6. I am not particularly in favour of the authors TTA approach. In this approach, the authors use both the forward and reverse directions of a reaction to essentially generate "more" data in a somewhat artificial way. The result may be that the prediction becomes directionally invariant but I believe there is a significant risk of overfitting in doing so. My preference would be for directional invariance to be "baked in" to the model. I am not, however, asking the author to develop a new model but I would demand to see more training statistics. A training curve is an important element of any ML paper that is sorely lacking in this work.

I look forward to reviewing a revised edition of this manuscript and I thank both the editor and author for the opportunity to be involved.

Best wishes,
Jason

Reviewer #3 (Remarks to the Author):

The present work by Choi presents a neural model for the prediction of transition state initial guess geometries for use in subsequent geometry optimization. These models address a key bottleneck in automated kinetic workflows, as traditional methods to generate TS geometries are both time consuming and failure-prone. Choi builds a model using transformer layers and gated recurrent units and proficiently takes advantage of model ensembles to predict the TS interatomic distance matrix. The predicted distance matrix is then used in a nonlinear optimization to recover the TS Cartesian coordinates. Evaluating the model on a dataset of general, small molecule organic reactions, the model achieves good convergence with guesses optimized at a high success rate and shows superior performance compared to existing ML models. My primary concerns with the manuscript are regarding clarity of the methods and analysis along with a broader discussion highlighting the limitations of the model. Hence, I recommend minor revisions to this manuscript before publication.

1. Kinetic modeling is a visual science, but the reader sees very few examples of the model's predictions. I recommend an additional figure, either in the main text or the supplement, showing examples of the model's predictions--both good and poor. Including both unimolecular and bimolecular reactions would be nice. This will help the reader gain confidence in the model's ability and may highlight areas for improvement.

2. I especially like the modeling choice to predict a multiplier to the ratio of the interpolated distance matrix and the TS distance matrix. Did you test predicting the distance value directly? I'm curious to know if this makes a substantial difference (not a necessary experiment).

3. I'm not sure that TTA is worth highlighting, since according to Fig. 2, it does not improve model performance much.

4. I appreciate the information Fig. 3b is trying to convey, but it's difficult to read, especially with all the different colored histograms in the background. At the very least, I suggest making this figure much larger, but I would try to improve its readability.

5. While it is nice that ~94% of the optimizations converged, one should perform frequency calculations and verify the presence of exactly one imaginary frequency to correctly characterize the saddle point. I highly recommend that this analysis be performed for all of the successful optimizations. They should definitely be performed for the 126 cases with energy differences higher than 0.1 kcal/mol. Without a proper frequency analysis, it is uncertain whether the structures are true TS structures (even if they were verified by IRC).

6. There are few key limitations the author should highlight. First, that the reactant and product

geometries used to generate TS guesses in this study were obtained from the Grambow dataset and are hence QM-optimized reactants and products. Thus, the model still requires QM-optimized reactants and products as input, which should be clearly stated in the manuscript. Second and more importantly, for any bimolecular reactions, the reactants and products need to be aligned in a reacting configuration, which is inherently the case in the Grambow dataset because the data were generated using the growing string method. A typical workflow performs opt+freq calculations on each species independently as traditional TST calculations expect partition functions for each species rather than for the VDW complex that is found in the Grambow dataset. For reactions that have 2 or 3 products, which represent about 30% of this dataset, it would normally require another step to align these species to create the complex that was used to train the model presented here. Without a robust alignment scheme, it is unclear whether or not the presented model can be successfully applied to bimolecular reactions. This limitation should also be clearly stated in the paper.

7. To train the TS model, Choi uses random splits on the Grambow dataset. However, several recent publications have noted the issues with using random splits on this dataset (see ref. 1 and 2). While the dataset contains ~12k unique reactions, it only contains ~1-2k unique reactants. So, when using a random split, some of the reactants may be duplicated in the test set. For TS geometry prediction, this is not a severe issue, but I suggest reproducing Fig. 2 using reactant-based scaffold splits (see ref. 2 for additional details). There's no need to redo the QM optimization section.

8. I didn't quite understand the model loss functions for the reactant and product readout layers. These layers are predicting the energy and entropy of the reactants and products? This description (pg. 14) could be clarified. Further, while I understand the motivation for these readout layers (i.e. benefits of multitask training), it is not demonstrated whether this strategy is actually useful. Experiments ablating these readout layers would be helpful.

9. The statement at the beginning of section 4.5 is not exactly true. First, the model from ref. 3 does in fact predict an interatomic distance matrix, but it reconstructs Cartesian coordinates from the predicted distance matrix in an end-to-end fashion. So the statement that "previous studies...directly infer the 3D atomic positions" isn't strictly true. In fact, the nonlinear optimization used here is nearly identical to the one used in ref. 3, aside from the fact that ref. 3 used learned weights (w_{ij}). Second, an important work which was missed in the citations (ref. 4), uses a related approach of predicting a Columb matrix intermediate and obtaining the Cartesian coordinates through an optimization.

10. The choice of 90 models for the ensemble seems a bit excessive and may negatively impact downstream usage (i.e. longer runtimes). It would be nice to include some analysis on how many models are truly necessary as the choice of 90 seems arbitrary.

[1].Heid, E., & Green, W. H. (2021). Machine learning of reaction properties via learned representations of the condensed graph of reaction. *Journal of chemical information and modeling*.

[2].Spiekermann, Kevin A., Lagnajit Pattanaik, and William H. Green. "Fast Predictions of Reaction Barrier Heights: Toward Coupled-Cluster Accuracy." *The Journal of Physical Chemistry A* 126.25 (2022): 3976-3986.

[3].Pattanaik, L., Ingraham, J. B., Grambow, C. A., & Green, W. H. (2020). Generating transition states of isomerization reactions with deep learning. *Physical Chemistry Chemical Physics*, 22(41), 23618-23626.

[4].Makoś, M. Z., Verma, N., Larson, E. C., Freindorf, M., & Kraka, E. (2021). Generative adversarial networks for transition state geometry prediction. *The Journal of Chemical Physics*, 155(2), 024116.

Reviewer #1 (Remarks to the Author):

This paper presents novel machine-learning approach for prediction of transition state structure optimization for non-periodic molecular reactions. And this outperforms previously suggested models which is quite promising. However there are several points which should be addressed prior to publication.

1. This work exploits extensive reaction dataset of Grambow et al. which contains 10k reactions with atomic indices already mapped between reactant and product. If I understood correctly, in Pair sequence interface layer process, permutation-invariance and size-extensivity are preserved in transformer encoder but it seems correct atom mapping of reactant and product seem to be assumed in "Bidirectional GRU" step. However in other many cases, outside of this dataset, this correct atom mapping between reactant and product is not guaranteed to exist and itself is not a trivial problem. Although this model exhibits excellent performance within current dataset as training input, the author may need to address its applicability toward general molecular reactions. Please clarify in the manuscript if this model can be also generally applicable to any elementary reaction or it requires the atom-mapping process as preliminary step.

➔ As the reviewer correctly pointed out, the pair sequence interaction layer requires atomic mapping information due to GRU operations which update pair features by adopting features of the same atomic pairs on different structures.

For a reaction database, because the structures undergo changes of connectivity, it is difficult to derive a quick solution of the atom-mapping problem. However, thanks to several advances in graph theory and linear programming methods, for small and medium size of molecules, solutions of atom mapping in reaction become more accessible. (Ref. 40&41 in the revised manuscript) Therefore, I hope this model will be able to be utilized for the reaction data that does not contain the atom mapping information. Regardless of my expectation, the limit of the model should be clearly described in the manuscript. Thus, to note this fact, I revised the 3rd paragraph of Section 4.2 as follows:

"Initially, the PSI layer updates pair features by reflecting other pairs in the same structure. This procedure utilizes a transformer encoder, which is a permutation-invariant and size-extensive operation. This first update is not affected by the pair features of other structures. Subsequently, atomic features are updated again to reflect the information from other structures. Because this second update proceeds

only on atomic pairs sharing the same index, the permutation invariance and size extensivity of the first dimension are still preserved. For the second update, a bidirectional GRU is used. Because a GRU layer must update the features of the same atomic pairs in three structures, atom mapping information is required to implement PSI layers. In this work, the atom mapping information contained in the databases was utilized. To apply PSI layers to data that do not have atom mapping information, such data must be obtained preliminary using various atom mapping methods.[40, 41]”

2. As this is being machine learning approach, its performance/accuracy scales with the size of the training set. Although dataset of Grambow et al. has vast number of reactions, real applications always involve building up their own reaction data and this is even more stipulated by the failure case given in this manuscript. In this work training-validation-test set as splited into 8-1-1 ratio, but would be interesting to present performance improvement with respect to number of training set (learning curve). Then readers might be able to expect approximated accuracy when trained with this model.

→ I agree that the performance dependency of ML model on the number of training data is important information for the potential user. The performance measurements with a different number of train datasets are performed and their results are presented in Figure S3 of the revised supporting information. The ensemble consisted of the model trained with 25% of the original training set (2392 reactions) yields 4.00% and 12.2pm for molecular MAPE and molecular MAE, respectively. Those errors are still much lower than the comparison models' errors. For authors who have the interest to apply the proposed model to an extremely small and focused database, the results of S_N2 reaction database which has 48 and 5 reactions for training and validation may be a reasonable reference. To convey additional test results, I revised the last paragraph of the 7th page as follows:

“To investigate the applicability of the proposed approach to a small database, model performances trained with S_N2 reaction database and reduced Grambow's database are measured. The S_N2 reaction database was published by the authors of TSNet.[25] For S_N2 reactions, the proposed ensemble approach yielded a molecular MAPE of 1.738% and molecular MAE of 4.54 pm. The best-reported MAE for the TSNet model is 18.31 pm.[25] The definition of error in the TSNet model study is slightly different from the molecular MAE. If the same definition to the TSNet results is applied, then the error of the ensemble prediction is 4.97 pm. These results indicate that the proposed model is more than three

times as accurate as TSNet. Although the S_N2 database contains only 48 training data, the proposed ML approach still yields high accuracy. Additionally, for the tests based on the reduced training set of Grambow's reaction database, the proposed model achieved high accuracy. The accuracy of ML prediction according to the size of training set is presented in Figure S3 of the supporting information. The proposed model trained with only 25% of randomly sampled reactions from the training set outperforms the comparison models trained with the entire training set. These test results verify that the proposed ML architecture can learn the interatomic distances of TS structures efficiently, even with a small number of reaction data.”

3. Fig 2. should be improved with better visualization. Font is too small and color should be more differentiable.

→ I updated Figure 2 with a larger font size and different colors in the revised manuscript and revised the corresponding caption.

4. It's hard to follow what actually “test-time augmentation (TTA)” means in current manuscript without reading the references cited. The author might want to elaborate further in the manuscript regarding this part.

→ To convey the concept of TTA to a potential reader who is not familiar with ML, I supplemented the explanation by introducing a primary example of TTA (image cases). Additionally, related to the comment raised by the 2nd and 3rd reviewers, the fact that my TTA implementation is free from potential issues of data augmentations is further explained in the first paragraph of the 5th page as the following:

“For the test subset, the molecular MAPEs of the single model and ensemble were measured as 3.681% and 3.407%, respectively. The corresponding molecular MAE values are 11.56 pm and 10.70 pm. The error is reduced further by test-time augmentation (TTA), which utilizes the results of inferences of augmented test inputs to mitigate the variance of test inferences. For image data, flipped, rotated, and translated test images were used to enhance the quality of predictions. TTA can be implemented in many different ways depending on the methods used to augment data and merge inferences.[29,30] In this study, augmented data were obtained by reversing the directions of chemical reactions and the predicted interatomic distances from both original (forward) and reversed (backward) reactions are averaged. This not only enhances accuracy, but also eliminates the directional

dependence of TS structures, which is an important invariance. Because this augmentation was not applied during training, no problems associated with artificial data such as reduced generalization were introduced.”

5. In Fig. 4. There are success after/before optimization. If author meant success as convergence of saddle-point optimization, how can you confirm its success even before optimization? Also success is ambiguous terminology which can be better expressed with “converged”.

- ➔ The goal of Figure 4 (a) is to represent the results of saddle point optimization whose initial structure is from the ML inference. As the reviewer expected, “Success” means the convergence of saddle point optimization. The legend entries in Figure 4 (a) are updated as Converged (After Opt.), Converged (After Opt.), and Failed. I thank you for the kind suggestion to improve the clarity of the legend.
- ➔ Although the present ML model yield a highly accurate TS structure, the remained question – how can we confirm the convergence of TS optimization without performing simulation? – is not covered by the current status of my work. There are some uncertainty quantification methods to measure how much the prediction results are reliable. However, in my opinion, those approaches cannot be direct solutions to the reviewer’s question because the convergence of saddle-point optimization is not solely determined by an initial TS structure. Many conditions of saddle-point optimization (e.g. convergence criteria and the maximum number of iterations) may affect the convergence. Here, I used the same method and parameters for saddle-point optimizations to reference calculations but there might be saddle-point optimization conditions to yield a higher convergence ratio than mine. Even though the convergence ratio is not the ultimate factor to describe the quality of ML predictions, I address the numbers because it could be practical guidance for a potential reader to roughly estimate how frequently manual processes are required when this ML model results are used for initial structures of saddle-point calculations. To convey the meaning of convergence ratio, I supplement the meaning of quantum chemical validation in the first paragraph of Section 2.2 as follows:

“To validate prediction quality, saddle point optimizations were conducted using the predicted TS structures as initial structures. Among the 1196 test molecules, 1122 (93.8%) of the molecular structures successfully converged. Among the failed saddle-point optimizations, 60 failed because the maximum number of iterations for geometry relaxation

was exceeded, six failed because the self-consistent field failed to converge, and eight failed for other reasons. Unlike the reference calculations, the saddle-point calculations performed in this work used ML inferences as initial structures. Without any manual processing of initial structures, a high convergence ratio for saddle point optimizations was still obtained. Additionally, it does not necessarily indicate that the ML model failed if the initial structures from the model do not converge because convergence is affected not only by initial structures but also by various parameters and methods of optimization. In this study, the same conditions for saddle-point optimization used in the reference calculations were adopted (detailed descriptions of calculation parameters are provided in Section 4.1.)”

6. In Fig.4, there are obviously four colors contained in histogram but only three of them are explained. What are yellow bars? Also (c) contains three colors and only two of them are labeled. Generally font size of the graph label should be larger. Here color should be more contrasting to make it differentiable.

→ The bars in Figure 4 of the original manuscript have some transparency to precisely exhibit the changes of other bars in overlapped ranges. I think the yellow bars that the reviewer mentioned are the overlapped region of green and orange bars. Similarly, in Fig. 4 (c), two types of bars and their overlap are presented. To avoid this confusion, I replotted Figure 4 (a) and (c) in the revised manuscript.

7. In page 10, “challenging but demanding work” is awkward expression.

→ Sorry for the awkward expression. I rephrase the last paragraph of the 10th page as follows:

“Multiple TSs for a chemical reaction are frequently observed in general polyatomic systems. In a strict sense, it is extremely difficult to target the most stable TS structure precisely. However, practically, the most stable TS structure can be found by exploring multiple TSs. Therefore, the generation of multiple reaction pathways is essential for determining the most favorable reaction pathways. Furthermore, to design catalysts or retrosynthetic pathways, the exploration of many competitive reactions is frequently required. [31–33] To tackle this problem, the generation of multiple TS structures using fast and accurate ML inference is demonstrated with the aid of reactant and product sampling.”

8. Regarding saddle-point optimization, author should clarify which specific algorithm has been used in Qchem and which convergence criteria has been imposed in this work. (e.g. Minimum force)

- The maximum number of SCF cycles and optimization are both 100. Tolerances for gradient, displacement and energy of optimization are 10^{-4} , 4×10^{-4} and 3.3×10^{-7} , respectively. These values are identical to the conditions of reference calculations published by Grambow et al.. To clarify this computation options, I added the following explanation in the last paragraph of 4.1 section.

“For both Pysisphus and the Psi4 package, default options were employed. The saddle point calculations using the QChem package were performed with the same options used to derive the original database results. The maximum number of SCF cycles and optimization operations were both 100. The tolerances for the gradients, displacement, and energy of optimization were 10^{-4} , 4×10^{-4} , and 3.3×10^{-7} , respectively.”

9. Author indicates rotational and translation entropy have been used to encode reaction property, but its notations are wrong. I.e. S_vib and S_rot, which are vibrational and rotational entropy. Also author needs elaboration why only two components of entropy are used instead of three (e.g. vibrational, rotational and translational entropy)

- Sorry for the typo. In this work, only rotational and vibrational entropies are used. (not translational entropy) Generally speaking, the translational entropy of gas molecules is not much different according to the chemical structures. Therefore, in many studies, vibrational and rotational entropies are used for the descriptor of chemicals. In the revised manuscript, the typo is corrected as follow:

“The second readout layer computes the contribution of each atom to total energy, vibrational entropy, and rotational entropy. (E_i , $S_{(vib,i)}$, $S_{(rot,i)}$)”

10. There are several typos throughout the manuscript and these have to be revised before publication.

- Thanks for your comment. I additionally performed proofreading of the submitted manuscript.

Reviewer #2 (Remarks to the Author):

Thank you for the opportunity to review the manuscript by Choi regarding the automated identification of transition state (TS) structures using machine learning (ML).

In the interests of transparency, I'm happy to reveal my identity to the author as Prof. Jason Pearson of the University of Prince Edward Island. This is particularly relevant in this case since I am the corresponding author of a paper that was extensively cited in this manuscript. In particular, my group and I have developed parallel technology for identifying TS structures by ML and Choi has rigorously tested our methodology alongside their own.

Overall I find the work to be a compelling case for the use of ML technology in the typical workflow of a computational chemist interested in elucidating mechanistic and kinetic detail of chemical reactions. Though I must admit that I have not checked out the authors freely available code from their GitHub repository, I have read the paper with interest and studied their methodology carefully. Based on this assessment I can say that the work has been soundly prepared and I see no reason to contrast any of the authors conclusions. Furthermore, the prediction of TS structures is an incredibly challenging endeavour (evidenced by a long history of widely varying methods for doing so) yet is critical for a thorough understanding of virtually any chemical process. As such I must recommend that the manuscript be published.

That being said, I am not without a series of more specific comments (both positive and negative), several of which I would require the author to address in a revised manuscript.

1. It is unclear to me how the author plans to handle chirality with their proposed model. As they state (in many places) most clearly on page 16, "... the ML model predicts only interatomic distances". It is, of course, an inconvenient truth that a stereogenic carbon with 4 unique substituents will produce an equivalent distance matrix of all enantiomers. I cannot interpret how the authors subsequent non-linear optimization step would correct this oversight. So, I ask the author to address this. Specifically, can the model succeed in stereochemical TS predictions? And if so, how?

- ➔ Predicting the chirality of a molecule is a challenging topic in structure predictions. The ML model that predicts interatomic distances, including mine, cannot systematically handle optical isomerism. In the non-linear optimization step, atomic positions are updated to have a close distance matrix to the predicted one without any special consideration of chirality.

Therefore, in the process of nonlinear optimization, the chirality is determined based on the initial structure, the linear–interpolated structure in this work. Even if a linear–interpolated structure is used, the correct chirality of TS structure is not guaranteed. Therefore, in my study, chirality predictions are not statistically studied. In the visualized reactions of the revised manuscript and supporting information, both valid and invalid predictions of chirality are observed. Figure 5 (b) shows the case that the chirality is well obtained, however, Figure S1 (a) shows the opposite case. To convey this limit of the distance–based TS structure prediction model, I added the explanations for chirality predictions in the last paragraph of 5th page as follows:

*“The interatomic distances obtained from ML inferences are highly accurate. However, they do not directly correspond to reliable 3D TS structures. Fortunately, based on nonlinear optimization using the results of inferences, accurate molecular geometries whose errors in terms of interatomic distances are less than those of both single–model and ensemble results can be obtained. This indicates that the remaining error in the predicted distances can be mitigated by constraining the set of interatomic distances to satisfy the Euclidean condition. **Despite highly accurate results using nonlinear optimization, because enantiomers are not distinguishable in terms of interatomic distances, nonlinear optimization cannot guarantee the correct chirality of TS structures. The incorrect prediction of chirality is not considered by the error metrics adopted in this study. Therefore, incorrect chirality prediction is observed even in the lowest–error case (0.88% molecular MAPE and 2.28 pm molecular MAE) shown in Figure S1, which plots few best and worst prediction results. This chirality issue is a common limitation of the ML model based on interatomic distance.[24]**”*

2. Unfortunately the grammar of the manuscript needs a lot of work. I trust that the Springer/Nature editorial team can work with the author to revise. I spotted no less than 20 errors.

➔ Sorry for the incorrect expression. I performed additional proofreading of the submitted manuscript.

3. I was particularly impressed with the authors ensemble approach, which endows the model with the ability to find multiple TS structures between a particular pair of RC and PC. This is a particularly clever solution to the problem of PES complexity.

→ Thanks for your compliment.

4. I found it peculiar that exactly 1196 "test molecules" were selected from the Grambow database for further analysis by optimization (page 8). However, the database contains 11961 reactions (page 13). Is this just a strange coincidence or a typo? If only a coincidence, can the author clarify how the 1196 test molecules were selected from the larger 11961 set?

→ The total size of the reaction database is 11961 and I used 80–10–10 random splitting. Hence, the size of the test set is 1196. It is explained in the first paragraph of the method section.

5. Again, I was confused about the statement on page 9 regarding the error in absolute energy of the optimized structures. It isn't clear to me how one measures this error. I understand that a TS is predicted from the ML model and that is a "predicted" structure. Subsequently a user can optimize that predicted structure using a traditional saddle point optimization from first principles. Therefore an "error" can be determined from the difference in energies of these two structures. However, What structure does one compare to if they wish to assess the error of the "optimized" TS structure? Even the reference database used the same theoretical technique (namely, ω B97X–D3/def2–TZVP). I would ask that the author please elaborate on their explanation of "error" for optimized structures.

→ The energy errors of the optimized structures are measured by the comparisons of the energy of optimized structures to the corresponding reference value. In this study, saddle point optimization is performed with the identical options used to compute reference results. The reason why the difference between the reference and my calculations happen is the initial structures of the saddle point optimization. The reference optimization calculations were performed with the initial structures computed from low–level quantum chemical calculations which are still much more expensive than ML inferences. If my calculation and the reference calculation capture the same TS structure, no energy difference would not be observed. In most optimization results, only a chemically meaningless amount of differences remain. Those small errors may be originated from the tolerance of optimization. On the other hand, more than 0.1 kcal/mol errors are observed in some cases and those differences may be induced by the different optimized structures. I further investigate the chemical validity of those cases. To clarify the potential difference between the optimized structure to the reference

structure, I added explanations in the first paragraph of Section 2.2 as follows:

“To validate prediction quality, saddle point optimizations were conducted using the predicted TS structures as initial structures. Among the 1196 test molecules, 1122 (93.8%) of the molecular structures successfully converged. Among the failed saddle point optimizations, 60 failed because the maximum number of iterations for geometry relaxation was exceeded, six failed because the self-consistent field failed to converge, and eight failed for other reasons. Unlike the reference calculations, the saddle point calculations performed in this work used ML inferences as initial structures. Although all options of saddle point optimization except for the initial structure used the same values as the reference options, saddle point calculation may fail to converge or converge to the different structure to the reference one. (detailed options for quantum chemical calculation parameters are provided in Section 4.1.) Without any manual processing of initial structures, a high convergence ratio for saddle point optimizations was still obtained. Additionally, it does not necessarily indicate that the ML model failed if the initial structures from the model do not converge because convergence is affected not only by initial structures but also by various parameters and methods of optimization. ”

6. I am not particularly in favour of the authors TTA approach. In this approach, the authors use both the forward and reverse directions of a reaction to essentially generate "more" data in a somewhat artificial way. The result may be that the prediction becomes directionally invariant but I believe there is a significant risk of overfitting in doing so. My preference would be for directional invariance to be "baked in" to the model. I am not, however, asking the author to develop a new model but I would demand to see more training statistics. A training curve is an important element of any ML paper that is sorely lacking in this work.

- I agree with the reviewer's point that imposing directional invariance on ML model would be a better solution than acquiring it through post-processing of inference results. For that, instead of a conventional bidirectional GRU layer, a recurrent neural network that can conserve directional symmetry would be an alternative. (This is my suggestion and I haven't implemented it yet.)
- Despite the limitation of the current model, TTA can be an attractive solution to preserve directional symmetry because, unlike conventional data augmentation methods, TTA is free from overfitting. The augmented

data are used in only test time which means no artificially generated reaction data is used in training. Furthermore, reversing the direction of reactions does not induce any noise of output results because a TS structure of the backward reaction is identical to the forward one. To adopt the reviewer's comment as well as other TTA-related comments from other reviewers, I revised explanation for TTA

“For the test subset, the molecular MAPEs of the single model and ensemble were measured as 3.681% and 3.407%, respectively. The corresponding molecular MAE values are 11.56 pm and 10.70 pm. The error is reduced further by test-time augmentation (TTA), which utilizes the results of inferences of augmented test inputs to mitigate the variance of test inferences. For image data, flipped, rotated, and translated test images were used to enhance the quality of predictions. TTA can be implemented in many different ways depending on the methods used to augment data and merge inferences.[29,30] In this study, augmented data were obtained by reversing the directions of chemical reactions and the predicted interatomic distances from both original (forward) and reversed (backward) reactions are averaged. This not only enhances accuracy, but also eliminates the directional dependence of TS structures, which is an important invariance. Because this augmentation was not applied during training, no problems associated with artificial data such as reduced generalization were introduced.”

Reviewer #3 (Remarks to the Author):

The present work by Choi presents a neural model for the prediction of transition state initial guess geometries for use in subsequent geometry optimization. These models address a key bottleneck in automated kinetic workflows, as traditional methods to generate TS geometries are both time consuming and failure-prone. Choi builds a model using transformer layers and gated recurrent units and proficiently takes advantage of model ensembles to predict the TS interatomic distance matrix. The predicted distance matrix is then used in a nonlinear optimization to recover the TS Cartesian coordinates. Evaluating the model on a dataset of general, small molecule organic reactions, the model achieves good convergence with guesses optimized at a high success rate and shows superior performance compared to existing ML models. My primary concerns with the manuscript are regarding clarity of the methods and analysis along with a broader discussion highlighting the limitations of the model.

Hence, I recommend minor revisions to this manuscript before publication.

1. Kinetic modeling is a visual science, but the reader sees very few examples of the model's predictions. I recommend an additional figure, either in the main text or the supplement, showing examples of the model's predictions—both good and poor. Including both unimolecular and bimolecular reactions would be nice. This will help the reader gain confidence in the model's ability and may highlight areas for improvement.

- ➔ To visualize more ML results, two types of extreme cases according to molecular MAPE and activation energy differences are additionally presented in the revised supporting information. The added visual contents cover both unimolecular and bimolecular reactions. Figure S1 plots three best and worst prediction cases in terms of molecular MAPE. It clearly shows the characteristics of predictions. (e.g. no systematic handling in chirality) Additionally, in Figure S5 and Figure S6, I enumerate some positive and negative error cases in terms of activation energy, which are the extended version of Fig. 5. The explanations corresponding to those additional visual contents are added as follows:

“The interatomic distances obtained from ML inferences are highly accurate. However, they do not directly correspond to reliable 3D TS structures. Fortunately, based on nonlinear optimization using the results of inferences, accurate molecular geometries whose errors in terms of interatomic distances are less than those of both single-model and ensemble results can be obtained. This indicates that the remaining error in the predicted distances can be mitigated by constraining the set of

interatomic distances to satisfy the Euclidean condition. Despite highly accurate results using nonlinear optimization, because enantiomers are not distinguishable in terms of interatomic distances, nonlinear optimization cannot guarantee the correct chirality of TS structures. The incorrect prediction of chirality is not considered by the error metrics adopted in this study. Therefore, incorrect chirality prediction is observed even in the lowest-error case (0.88% molecular MAPE and 2.28 pm molecular MAE) shown in Figure S1, which plots few best and worst prediction results. This chirality issue is a common limitation of the ML model based on interatomic distance.[24]” (The last paragraph in 5th page)

“Figure 5 presents two extreme IRC success cases in terms of their energy errors. The upper and lower subplots show the reference, predicted, and optimized TS structures for the cases with the most positive and most negative energy errors, respectively. More examples of large positive and negative error cases are plotted in Figure S5 and Figure S6, respectively. In Figure 5 as well as Figure S5 and Figure S6, one can see that the predicted TS structures are not significantly changed by saddle point optimization, whereas the reference TS structures are noticeably different from both the predicted and optimized TS structures. ……” (The second last paragraph on the 10th page)

2. I especially like the modeling choice to predict a multiplier to the ratio of the interpolated distance matrix and the TS distance matrix. Did you test predicting the distance value directly? I’m curious to know if this makes a substantial difference (not a necessary experiment).

➔ I additionally performed the training of the model targeting the interatomic distance of TS structure directly. I had never trained such a model but I had expected that the change of predicted target degrades the performance because the normalization of interatomic distances using a linear-interpolated structure significantly reduces the variance of target values. Surprisingly, the measured performance of the changed model is not much different from that of the proposed model. I did not further investigate the results due to the heavy computational cost of the quantum chemical validation. However, it can imply that the overwhelming performance of the proposed model originates from not the normalization of target values but its architecture. In this work, the changed model is not addressed in the proposed manuscript. In near future, I will prepare the manuscript for a more systematic investigation

of the normalization effect in TS structure predictions. Thank you for your insight.

3. I'm not sure that TTA is worth highlighting, since according to Fig. 2, it does not improve model performance much.

As the reviewer mentioned, TTA does not induce meaningful performance enhancement, especially in the ensemble case but, as I explained in the manuscript, TTA can provide invariant results according to the directions of reactions, which enhances the chemical justification of predictions. Therefore, TTA can generate more chemically reasonable results. To properly convey the meaning of TTA in this work and adopt comments raised by you and other reviewers (4th comment from 1st reviewer and 6th comment from 2nd reviewer), I supplement the explanation for TTA and its significance as follows:

“For the test subset, the molecular MAPEs of the single model and ensemble were measured as 3.681% and 3.407%, respectively. The corresponding molecular MAE values are 11.56 pm and 10.70 pm. The error is reduced further by test–time augmentation (TTA), which utilizes the results of inferences of augmented test inputs to mitigate the variance of test inferences. For image data, flipped, rotated, and translated test images were used to enhance the quality of predictions. TTA can be implemented in many different ways depending on the methods used to augment data and merge inferences.[29,30] In this study, augmented data were obtained by reversing the directions of chemical reactions and the predicted interatomic distances from both original (forward) and reversed (backward) reactions are averaged. This not only enhances accuracy, but also eliminates the directional dependence of TS structures, which is an important invariance. Because this augmentation was not applied during training, no problems associated with artificial data such as reduced generalization were introduced.”

4. I appreciate the information Fig. 3b is trying to convey, but it's difficult to read, especially with all the different colored histograms in the background. At the very least, I suggest making this figure much larger, but I would try to improve its readability.

→ Thanks for your kind suggestion on the visual content. To improve the readability of Fig. 3(b), bond types whose bonds are rarely distributed in the training set are excluded in visualization. In addition to that, the figure is slightly extended by the rearrangement of subplots. Instead of eliminating the distributions of all types of atomic pairs in the main manuscript, I added Figure S3 in the supplementary information to visualize the overall distribution. To note these changes, the caption of Figure 3 (b) is updated as follows:

(b) Absolute percentage errors and numbers of atomic pairs with different element sets according to interatomic distance. The dotted, dashed, and solid lines represent the average errors of TSNet, TSGen and the ensemble approach, respectively. The bars represent the frequencies of distances in the training set by atomic number. The red line represents a value of 156.6 pm, which is the criterion for the presence of a chemical bond. For clarity, only the distributions of selected atomic pairs are visualized. The distributions of the atomic pairs that are not visualized are plotted in Figure S2 in the supplementary information.

5. While it is nice that ~94% of the optimizations converged, one should perform frequency calculations and verify the presence of exactly one imaginary frequency to correctly characterize the saddle point. I highly recommend that this analysis be performed for all of the successful optimizations. They should definitely be performed for the 126 cases with energies differences higher than 0.1 kcal/mol. Without a proper frequency analysis, it is uncertain whether the structures are true TS structures (even if they were verified by IRC).

→ As the reviewer mentioned, a TS structure has one single negative (imaginary) frequency. In the previous manuscript, the number of imaginary frequencies of the converged structures was not addressed. To convey the results of frequency calculations, I added one paragraph to explain the frequency calculation results of all the optimized structures and the implication of a high success ratio in Section 2.2 of the revised manuscript as follows:

“For the converged structures, frequency calculations were performed and it was observed that 956 structures (80% of the 1196 test set data) had one negative frequency. The high success ratio is not direct evidence of high accuracy of the ML inferences but considering the fact that Grambow's reaction database includes many non-trivial reactions,

an 80% success ratio without any manual handling is a noteworthy achievement that reflects the fidelity of ML inferences as initial structures for saddle point calculations.

- Additionally, in the discussion of IRC analysis, I noted the number of IRC success structures having one imaginary frequency. Also, I added Figure S4 that shows distributions of structures having different numbers of imaginary frequencies To note that, I revised the manuscript as follows:

“In addition to frequency calculations, intrinsic reaction coordination (IRC) calculations were performed to investigate the validity of the optimized TS structures. The structures with errors greater than 0.1 kcal/mol (shown on the right side of the red line in Figure 4(b)) were selected for testing (126 cases). In principle, a TS structure has a single negative frequency and an IRC calculation beginning from a TS structure provides target reactants and products. However, among the 126 converged TS structures, there were only 45 structures that satisfied both conditions (8 and 58 structures did not satisfy the negative frequency and IRC conditions, respectively, and 15 structures failed on both criteria). If the reference reactant and product structures were obtained by IRC calculations, then the cases were classified into the IRC success group. Otherwise, they were classified into the IRC failure group. The overall distributions of both groups according to the energy error ($|E^{\text{Opt}} - E^{\text{Ref}}|$) are represented in Figure 4(c). Additionally, each distribution of structures having a different number of negative frequencies is plotted in Figure S4.”

6. There are few key limitations the author should highlight. First, that the reactant and product geometries used to generate TS guesses in this study were obtained from the Grambow dataset and are hence QM-optimized reactants and products. Thus, the model still requires QM-optimized reactants and products as input, which should be clearly stated in the manuscript. Second and more importantly, for any bimolecular reactions, the reactants and products need to be aligned in a reacting configuration, which is inherently the case in the Grambow dataset because the data were generated using the growing string method. A typical workflow performs opt+freq calculations on each species independently as traditional TST calculations expect partition functions for each species rather than for the VDW complex that is found in the Grambow dataset. For reactions that have 2 or 3 products, which represent about 30% of this dataset, it would normally require another step to align these species to create the complex that was used to train the model presented here. Without a robust alignment scheme, it is unclear whether or not the presented

model can be successfully applied to bimolecular reactions. This limitation should also be clearly stated in the paper.

- I agree that it is necessary to clearly state the limitations of the proposed idea. As the reviewer pointed out, the proposed ML model relies on the results of quantum chemical calculations because all data used in this study are from QM optimizations. Therefore, the proposed model does not include the sampling procedure of conformations for reactants and products. One example that I present in this manuscript is that the trained model can work with off-equilibrium geometries. However, it still requires opt+freq calculations because the sampling is performed based on normal mode sampling. For bi- and tri-molecular reactions, the proper selection of VDW complexes is not simply achieved by a single optimization. Therefore, typical sampling method and quantum chemical calculations are still unavoidable in the process of predicting TS structures despite the overwhelming performance of ML predictions. To convey this limitation of the current ML model and the need for sampling methods based on quantum chemical calculations, I revised the last paragraph of Section 3 as follows:

“To demonstrate the potential usage of the proposed model for searching multiple reaction paths, an autonomous reaction path finder was implemented and tested on a polyatomic reaction. By performing ML inference on the off-equilibrium conformations of reactants and products, and further refinement through saddle point calculations, various chemically meaningful TS structures were identified. This automated TS finder has huge potential to leverage ML applications for chemical reactions because it can be directly utilized for not only chemical applications (e.g., design of synthetic routes and catalysts), but also various ML applications related to chemical reactions (e.g., active learning). Although off-equilibrium conformations are utilized to predict TS structures, they are still sampled from well-optimized and aligned structures, which are obtained from quantum chemical calculations. This process is not trivial, particularly for bimolecular and trimolecular reactions, because the initial relative pose of molecules is a critical factor in determining reaction profiles. Therefore, to develop a more rigorous and realistic strategy for TS structure prediction, combinations of various ML approaches presented in this work (e.g., TTA and ensemble), as well as high-end quantum chemical calculation methods for configuration sampling are required.”

7. To train the TS model, Choi uses random splits on the Grambow dataset. However, several recent publications have noted the issues with using random splits on this dataset (see ref. 1 and 2). While the dataset contains ~12k unique reactions, it only contains ~1–2k unique reactants. So, when using a random split, some of the reactants may be duplicated in the test set. For TS geometry prediction, this is not a severe issue, but I suggest reproducing Fig. 2 using reactant-based scaffold splits (see ref. 2 for additional details). There's no need to redo the QM optimization section.

→ I trained the same ML model with the data from reactant-based scaffold splitting and summarized the results in Table S1 in the revised supporting information. In both MAPE and MAE metrics, the scaffold splitting results slightly outperform the random splitting results. It indicates that duplicates of the reactant's scaffold are not critical in learning TS structure as the reviewer mentioned. I added discussion in Section 4.1 of the revised manuscript as follows:

“Random sampling is the simplest method for splitting a database into train, validation, and test subsets. However, in a reaction database, random sampling may lead to the same structures being included in both the training and test sets because reactions sharing a reactant or product are treated as different reactions in a reaction database. To avoid duplicates of structures, in some ML applications for chemical reactions, reactant-based scaffold splitting, which assigns reactants having the same scaffold to the same subset, is employed.[35,36] Table S1 summarizes the accuracy of ensemble predictions trained using Grambow's database with two different sampling methods. The results indicate that the proposed model provides reliable accuracy, regardless of the splitting method. Therefore, only random sampling was used for further quantum chemical validations.”

8. I didn't quite understand the model loss functions for the reactant and product readout layers. These layers are predicting the energy and entropy of the reactants and products? This description (pg. 14) could be clarified. Further, while I understand the motivation for these readout layers (i.e. benefits of multitask training), it is not demonstrated whether this strategy is actually useful. Experiments ablating these readout layers would be helpful.

→ The second readout layer of the proposed model illustrated as pink boxes in Figure 1 predicts atomic contributions of energy, and two entropy values. The predicted properties are used to evaluate the second part of loss, L_2 in Equation 5. It is originally designed to impose Hammond's postulate on the model. In the original manuscript, I showed the prediction accuracy of model trained with two different c' values (0 and 1). If c' is zero, L_2 and L_3 are not minimized during the training,

which means multi-label learning is not used. In the original supporting information, the accuracy of all models belonging to the ensemble is enumerated but it did not well convey the effect of multi-label learning. Therefore, the table in the supporting (Table S3 in the revised one) is added and related discussion is added in the second last paragraph of Section 4.2 as follows:

“……The predicted values from the second readout layers are utilized for multi-label learning, which is frequently used to improve the quality of ML inferences by deriving desired information from related information. [42,43] However, in this study, additional labels reduced the accuracy of the ML model (see Table S3 in the supporting information.) A detailed description of the loss function design is provided in Section 4.4.”

9. The statement at the beginning of section 4.5 is not exactly true. First, the model from ref. 3 does in fact predict an interatomic distance matrix, but it reconstructs Cartesian coordinates from the predicted distance matrix in an end-to-end fashion. So the statement that “previous studies...directly infer the 3D atomic positions” isn't strictly true. In fact, the nonlinear optimization used here is nearly identical to the one used in ref. 3, aside from the fact that ref. 3 used learned weights (w_{ij}). Second, an important work which was missed in the citations (ref. 4), uses a related approach of predicting a Coulomb matrix intermediate and obtaining the Cartesian coordinates through an optimization.

➔ Sorry for the incorrect description of previous studies. As the reviewer correctly described, the ML model in Reference 3 (Ref[24] in the revised manuscript.) predicts interatomic distances and subsequently reconstructs atomic positions although, unlike my model, nonlinear optimization in TSGen is a part of ML predictions and its gradients are tracked in the backpropagation procedure. On the other hand, some TS prediction models do not rely on subsequent optimization. To clarify why some models use it and others do not, the first few sentences of Section 4.5 is revised as follows:

“An ML model such as TSNet directly outputs atomic positions satisfying invariance conditions.[25] However, in some studies, including this study, to preserve the constraints of atomic positions, ML models predict structure-dependent quantities (e.g., a distance matrix[24] or Coulomb matrix[45]) and atomic positions are reconstructed through a subsequent nonlinear optimization. In this study, similar to the TSGen model, a distance matrix was used as an optimization target. In TSGen, nonlinear optimization is a component of ML inference, whereas, in this study, nonlinear optimization was performed after the completion of ML inference. Therefore, the results of multiple ML models can be used in a

single nonlinear optimization together. From the initial atomic positions, X , the nonlinear optimization finds optimal positions, X^ , to minimize the differences relative to the ML predictions as follows:*

10. The choice of 90 models for the ensemble seems a bit excessive and may negatively impact downstream usage (i.e. longer runtimes). It would be nice to include some analysis on how many models are truly necessary as the choice of 90 seems arbitrary.

- The performance according to ensemble size is important information because, as the reviewer mentioned, the elapsed time for inference is linearly proportional to the size of the ensemble. To select the optimal ensemble size, the accuracy of ensemble predictions according to its size is additionally investigated. It is observed that the ensemble with 90 models is not the best case. In both molecular MAPE and molecular MAE, the ensemble with models sharing the best hyperparameters ($c=2000$ and $c'=0$) shows better performance than the 90 model case. To convey the reviewer's comment and performance of different ensemble cases, Table S3 is added in supporting information and the last paragraph of Section 4.5 is revised as follows:

“For ensemble predictions, multiple interatomic distance matrices are required prior to nonlinear optimization. Therefore, the computational costs of inferences are linearly dependent on the size of the ensemble, so the proper selection of an ensemble is important. Sets of models that were obtained from training using different seeds, hyperparameters, and training epochs were utilized. Six different combinations of hyperparameters (c and c' in Equation 5) are used. For each hyperparameter combination, three independent training runs with different random seeds were performed. During each training run, the model parameters in the epoch yielding the top five molecular MAPE values for the validation dataset were saved and used for the ensemble.

Ensemble predictions attempt to mitigate the variance of single-model predictions. However, the participation of low-accuracy models in an ensemble may reduce overall ensemble accuracy. The performances of all individual models are summarized in Table S4 and Table S3 summarizes the prediction accuracies of many different ensembles. For quantum chemical validation, ensemble predictions from 90 trained models were used and one can see that only the ensemble containing models trained using the optimal hyperparameter set ($c=2000$ and $c'=0$) outperforms the ensemble with 90 models.”

REVIEWER COMMENTS

Reviewer #1 (Remarks to the Author):

The author has properly addressed reviewer's comment but I'd like to point few things prior to publication.

1. Regarding comment #8, for convergence criteria author should provide corresponding units. Also What I asked for was to specify saddle point calculation algorithm (e.g. finite difference Davison procedure and associated solver convergence criterion). This is important, in my opinion, for reproducibility as there are various algorithms in QChem and author should provide exact information required for its potential reproduction.

Reviewer #2 (Remarks to the Author):

I have read the attached response letter in detail. Both to assess the authors response to my own comments and to review how they have responded to the other referees.

In all cases, I am satisfied that the manuscript can now be published.

Reviewer #3 (Remarks to the Author):

Thank you for responding to all of my comments and concerns. All of the additional changes to the manuscript improve its quality and make it suitable for publication as is. I'm quite impressed with the thoroughness of this rebuttal including the many additional experiments, figures appended to the SI, and expanded discussions. 80% success on frequency calculations seems a bit low, but the authors of the Grambow dataset did not perform IRCs on the dataset, so perhaps some of these failures are because of the underlying dataset rather than the model. I was not surprised to see that using 90 models is not the optimal setting, and using only two models with the best hyperparameters should speed up inference. I'm looking forward to the follow-up work! Well done!

Reviewer #1 (Remarks to the Author):

The author has properly addressed reviewer's comment but I'd like to point few things prior to publication.

→ I deeply thanks to you for thorough review. It was helpful to enhance readability of the manuscript.

1. Regarding comment #8, for convergence criteria author should provide corresponding units. Also What I asked for was to specify saddle point calculation algorithm (e.g. finite difference Davison procedure and associated solver convergence criterion). This is important, in my opinion, for reproducibility as there are various algorithms in QChem and author should provide exact information required for its potential reproduction.

→ All tolerance values are in atomic unit and the used method for TS optimization is finite difference Davidson procedure. I added these conditions on the revised manuscript.

Reviewer #2 (Remarks to the Author):

I have read the attached response letter in detail. Both to assess the authors response to my own comments and to review how they have responded to the other referees. In all cases, I am satisfied that the manuscript can now be published.

→ I deeply thanks to you for kind review and revealing yourself. I will continue the study on ML-based transition state predictions of chemical reactions. Please let me know any collaborations with me if you have interests.

Reviewer #3 (Remarks to the Author):

Thank you for responding to all of my comments and concerns. All of the additional changes to the manuscript improve its quality and make it suitable for publication as is. I'm quite impressed with the thoroughness of this rebuttal including the many additional experiments, figures appended to the SI, and expanded discussions. 80% success on frequency calculations seems a bit low, but the authors of the Grambow dataset did not perform IRCs on the

dataset, so perhaps some of these failures are because of the underlying dataset rather than the model. I was not surprised to see that using 90 models is not the optimal setting, and using only two models with the best hyperparameters should speed up inference. I'm looking forward to the follow-up work! Well done!

- ➔ I am truly grateful to you for your careful review and encourage. As you mentioned, frequency convergence is a bit low. Therefore, I will focus on combining other techniques from QM or chemical informatics to my model in near future.